# Learning Causal Models from Conditional Moment Restrictions by Importance Weighting

Masahiro Kato[1,2], Masaaki Imaizumi[2], Kenichiro McAlinn[3], Shota Yasui[1], and Haruo Kakehi[1]

[1]AI Lab, CyberAgent, Inc.
[2]The University of Tokyo
[3]Temple University

## Abstract

We consider learning causal relationships under conditional moment restrictions. Unlike causal inference under unconditional moment restrictions, conditional moment restrictions pose serious challenges for causal inference, especially in high-dimensional settings. To address this issue, we propose a method that transforms conditional moment restrictions to unconditional moment restrictions through importance weighting, using a conditional density ratio estimator. Using this transformation, we successfully estimate nonparametric functions defined under conditional moment restrictions. Our proposed framework is general and can be applied to a wide range of methods, including neural networks. We analyze the estimation error, providing theoretical support for our proposed method. In experiments, we confirm the soundness of our proposed method.

## 1 Introduction

Consider learning the causal relationship between airline ticket prices and demand. As one might expect, prices and demand rise and fall through the seasons, being affected by other events like vacation periods, which are called confounders and may or may not be observable. Due to confounders, naively inferring from this pattern that higher (lower) prices increase (decrease) demand would be incorrect, and potentially detrimental. Thus, controlling for confounding effects is essential. This issue frequently arises in practice, especially when learning causal (structural) relationships is essential to answer counterfactual questions regarding policy intervention and outcome (Hansen, 2022).

One approach to deal with confounding effects (like in the airline example above) is the instrumental variable (IV) approach (Wooldridge, 2002; Greene, 2003). In the IV approach, the conditional moment restriction is defined as such that the causal model satisfies the restriction of zero expected value given IVs, thus conditioning out the confounding effect. The simplest representation of this idea is the two-step least squares (2SLS) for linear models (Wooldridge, 2002; Greene, 2003). However, given the complex nature of the causal effect and confounding effect (and their relation), assuming a linear relation can be too strong. Thus, in this paper, we focus on nonparametric IV (NPIV) regressions, allowing for much more flexible estimation (Newey & Powell, 2003).

NPIV can be viewed as an instance of a more general framework of causal inference under conditional moment restrictions. In this light, the machinery for inference under conditional moment restrictions also applies to NPIV, as well as its shortcomings. One major issue with using the conditional moment restrictions for causal inference is that one must approximate the conditional expectation, which is often difficult to do (see, e.g., Newey (1993); Donald et al. (2003) for parametric and Newey & Powell (2003); Ai & Chen (2003) for nonparametric IVs using sieves). For instance, Lewbel (2007) and Otsu (2011) estimate the conditional expectation by local kernel density estimation. However, local kernel density estimation suffers under high dimensionality. For this problem, recent methods suggest the use of machine learning methods, such as neural networks (Hartford et al., 2017).

In this paper, we propose transforming conditional moment restrictions into unconditional moment restrictions by importance weighting using the conditional density ratio, which is defined as the ratio of the conditional probability density, conditioned on the IVs, to the unconditional probability density. We show that the unconditional expectation of a random variable weighted by the conditional density ratio is equal to the conditional expectation. Further, we show that it is possible to estimate the conditional density ratio with the least-squares method with a neural network. Once the conditional density ratio is estimated, the usual method of moments, such as GMM, can be used straightforwardly.

The contribution of this paper is as follows: (i) we propose a novel approach to convert conditional moment restrictions to unconditional moment restrictions by importance weighting; (ii) using our proposed transformation, we develop methods for NPIV; (iii) we analyze the estimation error

## 2 SETUP AND NOTATION

Among various problems of learning causal relationships from conditional moment restrictions, we focus on the NPIV regression for ease of discussion. Note that our proposed method can be applied to more general settings, similar to Ai & Chen (2003).

Suppose that the observations $\{(Y_i, X_i, Z_i)\}_{i=1}^n$ are i.i.d., where $Y_i \in \mathcal{Y} \subseteq \mathbb{R}$ is an observable scalar random variable, $X_i \in \mathcal{X} \subseteq \mathbb{R}^{d_X}$ is a $d_X$ dimensional explanatory variable, $Z_i \in \mathcal{Z} \subseteq \mathbb{R}^{d_Z}$ is a $d_Z$ dimensional random variable, called an IV, and $\mathcal{X}$ and $\mathcal{Z}$ are compact with nonempty interior. We also assume that the probability densities of $(Y_i, X_i, Z_i)$, $(Y_i, X_i)$, $Z_i$ exist and denote them by $p(y, x, z)$, $p(y, x)$, and $p(z)$, respectively. Let us define the causal relationships between $Y_i$ and $X_i$ as

$$Y_i = f^*(X_i) + \varepsilon_i,$$

where $f^* : \mathcal{X} \to \mathcal{Y}$ is a structural function, $\varepsilon_i$ is the sub-Gaussian error term with mean zero. To learn $f^*$, suppose the IV $Z_i$ satisfies the following conditional moment restrictions:

$$\mathbb{E}\left[\varepsilon_i | Z_i\right] = 0 \quad \forall i \in \{1, 2, \ldots, n\}. \tag{1}$$

Then, we also assume that under the conditional moment restriction, we can uniquely identify $f^*$. Our goal is to learn $f^*$ from the conditional moment restrictions in (1). If $Z_i = X_i$, this problem boils down to the estimation of the conditional expectation (regression function) $\mathbb{E}[Y_i|X_i]$. However, when $\mathbb{E}[\varepsilon_i|X_i] \neq 0$, $\mathbb{E}[Y_i|X_i]$ is not equivalent to $f^*(X_i)$; that is, typical regression analysis, such as least squares, may not return the correct estimate of $f^*(X_i)$.

## 3 PRELIMINARIES AND LITERATURE REVIEW

In this section, we briefly review causal inference under moment restrictions.

### 3.1 IV METHOD FOR LINEAR STRUCTURAL FUNCTIONS

One of the basic cases of using the IV is when $f^*$ is a linear model $X_i^\top \theta^*$ with $d_X$ dimensional vector $\theta^*$ and the error term $\varepsilon_i$ is correlated with the explanatory variable $X_i$. In this case, the parameter $\theta^*$ of the linear model can be estimated if there are IVs of dimension $d_X$ or more that satisfy the unconditional moment restrictions, $\mathbb{E}[Z_i \varepsilon_i] = \mathbf{0}_{d_Z}$, where $\mathbf{0}_d$ is a $d$ dimensional zero vector. In the just-identified case ($d_X = d_Z$), we can estimate $\theta^*$ as $\hat{\theta}_{\text{IV}} = \left(\frac{1}{n} \sum_{i=1}^n Z_i X_i^\top\right)^{-1} \frac{1}{n} \sum_{i=1}^n Z_i Y_i$. In the over-identified case ($d_X \leq d_Z$), we can estimate it by the 2SLS. In the 2SLS, we first regress $X_i$ by $Z_i$; then using a $d_X$ dimensional function $\hat{g}(Z_i)$ obtained from the first stage regression, we estimate $\theta^*$ as $\hat{\theta}_{\text{2SLS}} = \left(\frac{1}{n} \sum_{i=1}^n \hat{X}_i \hat{X}_i^\top\right)^{-1} \frac{1}{n} \sum_{i=1}^n \hat{X}_i Y_i$, where $\hat{X}_{d,i} = Z_i^\top \left(\frac{1}{n} \sum_{j=1}^n Z_j Z_j^\top\right)^{-1} \frac{1}{n} \sum_{j=1}^n Z_j X_{d,j}$ and $\hat{X}_i = (\hat{X}_{1,i}, \ldots, \hat{X}_{d_X,i})^\top$.

More generally, we can formulate the estimation of the linear structural function by unconditional moment restrictions defined using the IV; that is, $\mathbb{E}[m(\theta^*; Y_i, X_i, Z_i)] = \mathbf{0}_{d_m}$, where $\theta^* \in \Theta$ is a parameter representing the causal effect, $\Theta$ is the parameter space, and $m : \Theta \times \mathbb{R} \times \mathcal{X} \times \mathcal{Z} \to \mathbb{R}^{d_m}$ is a $d_m$ dimensional moment function. Then, the GMM estimator is defined as $\hat{\theta}_{\text{GMM}} = \arg\min_{\theta \in \Theta} \left(\frac{1}{n} \sum_{i=1}^n m(\theta; Y_i, X_i, Z_i)\right)^\top W_{d_m} \left(\frac{1}{n} \sum_{i=1}^n m(\theta; Y_i, X_i, Z_i)\right)$, where $W_{d_m}$ is a $d_m \times d_m$ weight matrix. If $W_{d_m}$ is chosen as $W_{d_m}^* \propto \mathbb{E}[m(\theta^*; Y_i, X_i, Z_i)^\top m(\theta^*; Y_i, X_i, Z_i)]$,

the estimator $\hat{\theta}_{\text{GMM}}$ is efficient under the posited moment conditions. Note that the GMM includes the 2SLS as a special case where $\varepsilon_i$ has the same variance among $i \in \{1, \ldots, n\}$ and the zero covariance.

Three methods have been proposed to obtain $W^*_{d_m}$; two-step GMM, iterative GMM (Hayashi, 2000), and continuous updating GMM (CU estimator; CUE, Hansen et al., 1996). These estimators have the same asymptotic distribution, but different non-asymptotic properties. In particular, CUE is known as a special case of the generalized empirical likelihood (GEL) estimator (Owen, 1988; Smith, 1997), which plays an important role in estimation with many moment restrictions (Newey & Smith, 2004).

## 3.2 NPIV AND LEARNING FROM CONDITIONAL MOMENT RESTRICTIONS

In linear structural functions, zero covariance between the IV and error term suffices for identification. However, in nonparametric structural functions, we require a stronger restriction: the error term has conditional expectation zero given the IVs. Then, we can characterize the solution of NPIV as that of an integral equation $K(z) = \mathbb{E}[Y_i | Z_i = z] = \mathbb{E}[f^*(X_i)|z] = \int f^*(x) dF(x|z)$ with $F$ denoting the conditional c.d.f of $x$ given $z$. The identification also results in the uniqueness of the solution.

Several estimators have been proposed. Newey & Powell (2003) proposes a nonparametric analogue of the 2SLS for linear structural functions. They first define the linear-in-parameter model as a model to approximate the nonparametric structural function $f^*$, where they use a linear approximation with basis expansion, such as sieve (series) regression. Let us denote the approximation as $f^*(X_i) \approx \psi(X_i)^\top \theta^*$, where $\psi : \mathcal{X} \to \mathbb{R}^{d_\psi}$, $d_\psi \leq n$ is a vector-valued function consisting of outputs of basis functions. Then, they conduct the 2SLS as follows: (i) they define a linear-in-parameter model as an approximation to the nonparametric model of $Z_i$, and estimate $\mathbb{E}[\psi(X_i)|Z_i]$ using that model; (ii) they regress $Y_i$ by $\mathbb{E}[\psi(X_i)|Z_i]$ to estimate $\theta^*$. In contrast, Ai & Chen (2003) proposes a nonparametric analogue of the GMM. Ai & Chen (2003) also approximate the nonparametric structural function $f^*$ by a linear-in-parameter model. However, unlike Newey & Powell (2003), Ai & Chen (2003) estimates $\mathbb{E}[(Y_i - f(X_i))|Z_i]$, instead of estimating $\mathbb{E}[\psi(X_i)|Z_i]$, by another linear-in-parameter model. Using the estimator of $\mathbb{E}[(Y_i - f(X_i))|Z_i]$, Ai & Chen (2003) transforms conditional moment restrictions into unconditional ones and apply the GMM to estimate $f^*$. In addition to these two typical methods, Darolles et al. (2011), Carrasco et al. (2007a) propose their own methods for NPIV.

This is how the NPIV problem is typically cast into the framework of conditional moment restrictions, where a parameter representing the causal relationship satisfies $\mathbb{E}[m(\theta^*; Y_i, X_i, Z_i)|Z_i] = \mathbf{0}_{d_m}$. As with NPIV, we often define the moment function as a function of the nonparametric function $f$ instead of the parameter $\theta$. We can estimate $f^*$ defined under conditional moment restrictions using variants of GMM or empirical likelihood, e.g., Ai & Chen (2003); Domínguez & Lobato (2004); Otsu (2011); Lewbel (2007), by transforming the conditional moment restrictions to unconditional ones.

## 3.3 APPROACHES IN THE MACHINE LEARNING LITERATURE

**2SLS with more flexible models.** Hartford et al. (2017) extends the two-stage approach by employing a neural network density estimator, and Singh et al. (2019) does it by conditional mean embedding in RKHS. For example, in Hartford et al. (2017), they first approximate $K(z)$ by estimating $F(x|z)$ with neural networks. Then, for $B$ samples $\{\tilde{X}_j\}_{j=1}^B$ generated from the estimator $\hat{F}(x|z)$, they train neural networks by minimizing the empirical risk $\frac{1}{n} \sum_{i=1}^n \left( Y_i - \frac{1}{B} \sum_{\tilde{X}_j \sim \hat{F}(X_j|Z_i)} f(\tilde{X}_j) \right)^2$.

In contrast, Xu et al. (2021a) first approximates $f(X_i)$ by neural networks and regard the last layer as $\psi(X_i)$ in the 2SLS of NPIV. Then, they predict $\psi(X_i)$ by another model $\varphi(Z_i)$. Because $\psi(X_i)$ is transitive in the learning process, they alternatively train the models $\varphi(Z_i)$ and $f(X_i)$.

**Minimax approach.** There has also been a recent surge in interest in minimax approaches that reformulate conditional moment restrictions as a minimax optimization problem (Bennett et al., 2019; Bennett & Kallus, 2020; Muandet et al., 2020; Chernozhukov et al., 2020; Liao et al., 2020). For this approach, Dikkala et al. (2020) shows a strong theoretical guarantee on the estimation error bound.

## 4 PROPOSED METHOD: NPIV BY IMPORTANCE WEIGHTING

Suppose that $p(y, x) > 0$ for all $(y, x) \in \mathcal{Y} \times \mathcal{X}$. Note that we have already assumed the existence of $p(y, x|z)$ for all $(y, x, z) \in \mathcal{Y} \times \mathcal{X} \times \mathcal{Z}$ by assuming the existence of the conditional moment restrictions. Define the conditional density ratio function $r : \mathcal{Y} \times \mathcal{X} \times \mathcal{Z} \to (0, C)$ as

$$r^*(y, x|z) = \frac{p(y, x|z)}{p(y, x)} = \frac{p(y, x, z)}{p(y, x)p(z)},$$

for $0 < C < \infty$. We assume the existence of the conditional density ratio function.

**Assumption 1.** *For all $(y, x, z) \in \mathcal{Y} \times \mathcal{X} \times \mathcal{Z}$, the conditional density ratio $r^*(y, x|z)$ exists.*

Then, we transform conditional moment restrictions into unconditional moment restrictions as

$$\mathbb{E}[(Y_i - f(X_i))|z] = \int (y - f(x)) \frac{p(y, x|z)}{p(y, x)} p(y, x) \mathrm{d}y \mathrm{d}x = \mathbb{E}\left[ (Y_i - f(X_i)) r^*(Y_i, X_i|z) \right].$$

Thus, if we know the conditional density ratio function $r^*(y, x|z)$, we can approximate $\mathbb{E}[(Y_i - f(X_i))|z]$ for $z \in \mathcal{Z}$ by the following sample average:

$$\frac{1}{n} \sum_{i=1}^{n} (Y_i - f(X_i)) r^*(Y_i, X_i|z).$$

Based on this property, we propose the following two-stage method: (i) estimate the conditional density ratio function $r^*$; (ii) approximate conditional moment restrictions by the sample average of $(Y_i - f(X_i))$ using the estimate of the conditional density ratio function $r^*$. Since we do not know the true value of the conditional density ratio $r^*$ and cannot calculate the expected value, we consider replacing $r^*$ with an estimator $\hat{r}$ and approximating the expected value with the sample mean.

### 4.1 CONDITIONAL DENSITY RATIO ESTIMATION

First, we consider estimating $r^*(y, x|z)$. While it is possible to estimate the probability density functions of the numerator and the denominator, individually, following Vapnik's principle (Vapnik, 1998), we should avoid solving more difficult intermediate problems. Sugiyama et al. (2012) summarizes various methods to estimate the density ratio directly. Inspired by *least-squares importance fitting* (LSIF) of Kanamori et al. (2009), we estimate the conditional density ratio by minimizing

$$\tilde{r} = \arg\min_{r \in \tilde{\mathcal{R}}} \frac{1}{2} \mathbb{E}_{Y,X} \left[ \mathbb{E}_Z \left[ \left( r^*(Y_i, X_i|Z_j) - r(Y_i, X_i|Z_j) \right)^2 \right] \right],$$

where $\tilde{\mathcal{R}}$ denotes a set of measurable functions and for a function $g : \mathcal{Y} \times \mathcal{X} \times \mathcal{Z} \to \mathbb{R}$, $\mathbb{E}_{Y,X}[\mathbb{E}_Z[g(Y_i, X_i, Z_j)]]$ denotes $\int g(y, x, z)p(y, x)p(z)\mathrm{d}y\mathrm{d}x\mathrm{d}z$. It is easy to confirm that $\tilde{r} = r^*$ by taking the first derivative of the risk. Because this risk includes the unknown $r^*$, it may seem intractable objective function. However, we can obtain the risk that does not include $r^*$ as

$$\arg\min_{r \in \tilde{\mathcal{R}}} \frac{1}{2} \left( \mathbb{E}_{Y,X} \left[ \mathbb{E}_Z \left[ r^{*2}(Y_i, X_i|Z_j) - 2r^*(Y_i, X_i|Z_j)r(Y_i, X_i|Z_j) + r^2(Y_i, X_i|Z_j) \right] \right] \right)$$

$$= \arg\min_{r \in \tilde{\mathcal{R}}} \left\{ -\mathbb{E}_{Y,X} \left[ \mathbb{E}_Z \left[ r^*(Y_i, X_i|Z_j)r(Y_i, X_i|Z_j) \right] \right] + \frac{1}{2} \mathbb{E}_{Y,X} \left[ \mathbb{E}_Z \left[ r^2(Y_i, X_i|Z_j) \right] \right] \right\}$$

$$= \arg\min_{r \in \tilde{\mathcal{R}}} \left\{ -\mathbb{E}_{Y,X,Z} \left[ r(Y_i, X_i|Z_i) \right] + \frac{1}{2} \mathbb{E}_{Y,X} \left[ \mathbb{E}_Z \left[ r^2(Y_i, X_i|Z_j) \right] \right] \right\}.$$

Here, we used $\mathbb{E}_{Y,X}[\mathbb{E}_Z[r^*(Y_i, X_i|Z_j)r(Y_i, X_i|Z_j)]] = \mathbb{E}_{Y,X}\left[\mathbb{E}_Z\left[\frac{p(Y_i, X_i, Z_j)}{p(Y_i, X_i)p(Z_j)} r(Y_i, X_i|Z_j)\right]\right] = \mathbb{E}_{Y,X,Z}[r(Y_i, X_i|Z_i)]$. For some hypothesis class $\mathcal{R}$, by approximating the risk with its sample approximation, we estimate the conditional density ratio as

$$\hat{r} = \arg\min_{r \in \mathcal{R}} \left\{ -\frac{1}{n} \sum_{i=1}^{n} r(Y_i, X_i|Z_i) + \frac{1}{2} \frac{1}{n} \sum_{j=1}^{n} \frac{1}{n} \sum_{i=1}^{n} r^2(Y_i, X_i|Z_j) \right\}.$$

For the hypothesis class $\mathcal{R}$, we can use various models, such as linear-in-parameter models and neural networks. Suzuki et al. (2008) also proposes a similar formulation based on maximum likelihood estimation with constraints, which is not easy to solve with neural networks.

## 4.2 NPIV REGRESSION BY IMPORTANCE WEIGHTING

If we know the conditional density ratio function $r^*$, we can obtain unconditional moment restrictions $\mathbb{E}_{Y,X}\left[\left(\rho(f;Y_i,X_i,Z_1,r^*)\,\rho(f;Y_i,X_i,Z_2,r^*)\,\cdots\,\rho(f;Y_i,X_i,Z_n,r^*)\right)^\top\right] = \mathbf{0}_n$, where $\rho(f;Y_i,X_i,z,r^*) = \left(Y_i - f(X_i)\right)r^*(Y_i,X_i|z)$. By replacing $r^*$ and its expectation with its estimator and the sample average, we obtain the sample vector moment restrictions, $\frac{1}{n}\sum_{i=1}^n\left(\rho(f;Y_i,X_i,Z_1,\hat{r})\,\rho(f;Y_i,X_i,Z_2,\hat{r})\,\cdots\,\rho(f;Y_i,X_i,Z_n,\hat{r})\right)^\top$.

Once we obtain the sample average, we can apply various methods for learning $f^*$ from the unconditional moment restriction. For instance, for a hypothesis class $\mathcal{F}$, we estimate $f^*$ by minimizing

$$\mathcal{R}_n(f,\hat{r}) = \frac{1}{n}\sum_{j=1}^n\left(\frac{1}{n}\sum_{i=1}^n(Y_i - f(X_i))\hat{r}(Y_i,X_i|Z_j)\right)^2. \tag{2}$$

This objective function is closely related with the projected mean squared error (MSE) introduced in Dikkala et al. (2020), defined as

$$\mathcal{L}(f,r^*) := \mathbb{E}_Z\left[\left(\mathbb{E}_{Y,X}\left[(f^*(X_i) - f(X_i))r^*(Y_i,X_i|Z_j)\right]\right)^2\right], \tag{3}$$

The objective function (2) is a special case of the GMM and GEL. In the GMM, we estimate $f^*$ by minimizing $\frac{1}{n}\sum_{j=1}^n\left(\frac{1}{n}\sum_{i=1}^n(Y_i - f(X_i))\hat{r}(Y_i,X_i|Z_j)\right)^\top w_j\left(\frac{1}{n}\sum_{i=1}^n(Y_i - f(X_i))\hat{r}(Y_i,X_i|Z_j)\right)$, where $w_j > 0$ is a weight (Ai & Chen, 2003). Thus, our framework of transforming conditional moment restrictions to unconditional moment restrictions using importance weighting allows us to conduct causal inference using NPIV with conditional moment restrictions using a variety of models, such as linear-in-parameter models with basis functions and neural networks.

**Linear-in-parameter models.** As an example, we introduce a linear-in-parameter model with some basis functions. Here, let us consider using the Gaussian kernel as the basis function. For $x \in \mathcal{X}$, let $\varphi(x;\sigma^2) = \left(K(x,X_1;\sigma^2),\ldots,K(x,X_n;\sigma^2)\right)^\top$, where $K(x,X_u;\sigma^2) = \exp\left(-\frac{\|x-X_u\|_2^2}{2\sigma^2}\right)$ is the Gaussian kernel with a hyperparameter $\sigma^2 > 0$ and $\|\cdot\|_2$ is the $L_2$ norm. Then, we define a linear-in-parameter model as $f(x;\sigma^2) = \beta^\top\varphi(x;\sigma^2) + \beta_0$, where $\beta \in \mathbb{R}^n$, and $\beta_0 \in \mathbb{R}$.

**Neural networks.** We can also use neural networks for approximating $f^*$. In this case, we need to carefully determine the network structures because we cannot uniquely determine the solution owing to the overparameterization. In existing studies, it is assumed that the network modes are well defined for the complexity of the nonparametric function $f^*$ and the sample size $n$.

**Advantages of our proposed method.** Our method has the following three advantages: (i) our method is applicable to general causal inference problems, and is not limited to the NPIV problem formulated in Section 2, as well as the Ai & Chen (2003), (ii) our method can deal with high dimensional variables, owing to the machine learning technique. This is in contrast to related studies Ai & Chen (2003) and Otsu (2011), which use a sieve and Nadaraya-Watson estimator, respectively, and hence do not work in high dimension settings, and (iii) our method is computationally efficient by the use of the importance weight approach. A similar study Hartford et al. (2017) requires additional sampling of $Y_i$ from an estimated conditional density $p(y|z,y)$. Another related study Dikkala et al. (2020) requires a difficult algorithm to solve its minimax optimization problem. Our importance weight approach avoids such computational burden.

## 4.3 LEARNING WITH OVERPARAMETRIZED MODELS

In learning from moment restrictions, the structural function $f^*$ is determined by the equations that the expected value of random variables satisfy. When approximating the structure function $f^*$ using a model with more parameters than the sample size, the solution cannot be uniquely determined. When minimizing the prediction error directly, such overparameterization may not pose a major problem, and, in fact, may be necessary to improve the generalization error, as a recent finding suggests in the case of linear regression (Bartlett et al., 2020). However, in NPIV, the objective function is

the set of conditional moment restrictions. Unlike in direct prediction error minimization, a model trained to minimize empirically approximated conditional moment restrictions does not necessarily minimize the MSE for $f^*$ ($\mathbb{E}[(f(X_i) - f^*(X_i))^2]$). In fact, we empirically confirm that methods using neural network sometimes do not work well partly because of their overfitting to empirical moment minimization, not to the empirical MSE for $f^*$. Despite these potential problems, there is a strong motivation to use neural networks, owing to the reported superiority in some applications, such as computer vision (Xu et al., 2021a;b; Yuan et al., 2021), natural language processing (Ash et al., 2019; Chen et al., 2020) tasks. For this reason, we introduce a heuristic to avoid this problem.

As we explained, the parameters are not uniquely determined by conditional moment restrictions alone. Therefore, from the set of parameters satisfying conditional moment restrictions, we need to select a set of parameters that works well in prediction. We consider training a model that has the minimum MSE with $Y_i$ while satisfying conditional moment restrictions as follows:

$$\min_{f \in \mathcal{F}} \mathbb{E}\big[(Y_i - f(X_i))^2\big] \quad \text{s.t. } \mathbb{E}\left[(Y_i - f(X_i))|Z_j\right] = 0 \quad \forall j \in \{1, 2, \ldots, n\}.$$

Since it is difficult to solve constrained optimization with neural networks, we propose to solve the penalized optimization. In the case of linear combination with penalties, we train the model by

$$\min_{f \in \mathcal{F}} \frac{1}{n} \sum_{i=1}^{n} (Y_i - f(X_i))^2 + \eta \sum_{j=1}^{n} \left( \frac{1}{n} \sum_{i=1}^{n} (Y_i - f(X_i))\hat{r}(Y_i, X_i|Z_j) \right)^2, \tag{4}$$

where $\eta \geq 0$ is a regularization coefficient.

The motivation of this heuristic is to select a function $f(x)$ that is the closest to $\mathbb{E}[Y_i|X_i = x]$, among multiple functions satisfying the conditional moment restriction. This heuristic works well when $f^*(x)$ takes a near value of $\mathbb{E}[Y_i|X_i = x]$ while $f^*(x) \neq \mathbb{E}[Y_i|X_i = x]$.

## 5 ESTIMATION ERROR ANALYSIS

We show the estimation error of the conditional density ratio $r^*$ and structural function $f^*$. We denote the sample counterpart of $\mathcal{L}(f, r^*)$ as

$$\mathcal{L}_n(f, r^*) = \frac{1}{n} \sum_{j=1}^{n} \left( \frac{1}{n} \sum_{i=1}^{n} (f^*(X_i) - f(X_i))r^*(Y_i, X_i|Z_j) \right)^2.$$

Let us denote the distributions of $(Y_i, X_i)$ and $Z_i$ by $P$ and $Q$, respectively, and define the $L^2$ risk of a function $g$ with $P$ and $Q$ as $\|g\|_{L^2(P \times Q)}^2 = \int \int g^2(w)\mathrm{d}P\mathrm{d}Q$. Let us denote the distribution of $(Y_i, X_i, Z_i)$ by $O$, and define the $L^2$ risk of a function $g$ with $O$ as $\|g\|_{L^2(O)}^2 = \int g^2(w)\mathrm{d}O$. We put the following assumptions on the error term $\varepsilon_i$.

**Assumption 2.** *The error term $\varepsilon_i$ is sub-Gaussian random variables. In addition, the distributions of $X_i$ and $Z_i$ have probability densities that are finite and bounded away from zero.*

Note that the randomness of $Y_i$ depends on $\varepsilon_i$ and $X_i$. Define the hypothesis classes of the conditional density ratio $r^*$ and structural function $f^*$ as $\mathcal{R}$ and $\mathcal{F}$, respectively. Suppose that the hypothesis classes are Vapnik–Chervonenkis (VC) class (for rigorous definition, see Section 2.6 in van der vaart & Wellner (1996)). This class include the true models, and the hypothesises are bounded.

**Assumption 3.** *The hypothesis class $\mathcal{R}$ is VC class, includes the true model, $r^* \in \mathcal{R}$, and all $r \in \mathcal{R}$ are uniformly bounded by $B > 0$.*

**Assumption 4.** *The hypothesis class $\mathcal{F}$ is VC class, includes the true model, $f^* \in \mathcal{F}$, and all $f \in \mathcal{R}$ are uniformly bounded by $B > 0$.*

We define a measure of complexity of the hypothesis classes: for $\mathcal{F}$, we define the complexity $\mathcal{C}_B(\mathcal{F}) := \int_0^B \sqrt{\log \mathcal{N}(\delta', \mathcal{F}, \|\cdot\|_{L^\infty})}\mathrm{d}\delta'$ with a covering number $\mathcal{N}(\delta', \mathcal{F}, \|\cdot\|) := \inf\{N|\{f_j\}_{j=1}^N \text{ s.t. } \mathcal{F} \subset \cup_{j=1}^N \{f|\|f - f_j\| \leq \delta'\}\}$ in terms of a sup-norm $\|f\|_{L^\infty} = \sup_x |f(x)|$.

In this following parts, for the conditional density ratio estimation, we derive the bound of the MSE $\|\hat{r} - r^*\|_{L^2(P \times Q)}$; for the structural function estimation, we derive the bound of the projected MSE $\mathcal{L}(\hat{f}, r^*)$, which corresponds to the upper bound of the MSE of $\|\hat{f} - f^*\|_{L^2(O)}$ (Section 5.3).

## 5.1 MSE OF THE CONDITIONAL DENSITY RATIO

First, we consider the MSE of the conditional density ratio. For a multilayer perception with ReLU activation function (Definition 1), we show the following bound. The proof is shown in Appendix B.2.

**Lemma 1** (MSE of $r^*$). *Suppose that Assumptions 1–3 hold. Let $I(r)$ be a non-negative function on $\mathbb{R}$ and $I(r^*) < \infty$. Define $\mathcal{R}_M = \{r \in \mathcal{R} : I(r) \leq M\}$ satisfying $\mathcal{R} = \bigcup_{M \geq 1} \mathcal{R}_M$. Suppose that there exist $c_0 > 0$ and $0 < \gamma < 2$ such that $\sup_{g \in \mathcal{R}_M} \|r - r^*\| \leq c_0 M$ and $\sup_{\substack{r \in \mathcal{R}_M \\ \|r - r^*\|_{L^2(P)} \leq \delta}} \|r - r^*\|_\infty = c_0 M$ for all $\delta > 0$, and that $\log \mathcal{N}(\delta, \mathcal{R}_M, \|\cdot\|_{L^\infty}) = \mathcal{O}\left(M/\delta\right)^\gamma$. Then,*

$$\|\hat{r} - r^*\|_{L^2(O)} = \mathcal{O}_p\left(n^{-1/(2+\gamma)}\right) \tag{5}$$

## 5.2 PROJECTED MSE OF STRUCTURAL FUNCTION $f^*$

Next, we consider bounding the projected MSE of the structural function $f^*$. To bound the projected MSE, we use a technique associated with U-statistics. We first obtain the following lemma. Let $(Y_i, X_i)$ be $W_i$, and $W_i'$ be the i.i.d. copy of $W_i$. The proof is shown in Appendix B.3.

**Lemma 2.** *Suppose that Assumptions 1–4 hold. For any $f \in \mathcal{F}$ and $r \in \mathcal{R}$, $\mathcal{L}(f, r) = \widetilde{\mathcal{L}}(f, r) + o_p(1)$ holds as a freely chosen hyper-parameter as $n \to \infty$, where*

$$\widetilde{\mathcal{L}}(f, r) = \mathbb{E}_{W,W',Z}\left[(f^*(X_i) - f(X_i))r(W_i|Z_i)(f^*(X_i') - f(X_i'))r(W_i'|Z_i)\right],$$

Regarding the form, we define an empirical version of $\widetilde{\mathcal{L}}(f, r)$ in an U-statistic form:

$$\widetilde{\mathcal{L}}_n(f, r) = \frac{1}{n} \sum_{j=1}^n \frac{1}{n(n-1)} \sum_{i,i'=1, i \neq i'}^n (f^*(X_i) - f(X_i))r(W_i|Z_j)(f^*(X_{i'}) - f(X_{i'}))r(W_{i'}|Z_j).$$

By a property of U-statistics (for example, see Arcones & Giné (1993)), $\mathcal{L}_n(f, r) = \widetilde{\mathcal{L}}_n(f, r) + \mathcal{O}_p(1/n)$ clearly holds. Then, we can decompose the projected MSE $\mathcal{L}(\hat{f}, r^*)$ as follows:

$$\mathcal{L}(\hat{f}, r^*) = \underbrace{\widetilde{\mathcal{L}}(\hat{f}, r^*) - \widetilde{\mathcal{L}}(\hat{f}, \hat{r})}_{=:\Delta_r} + \underbrace{\widetilde{\mathcal{L}}(\hat{f}, \hat{r}) - \widetilde{\mathcal{L}}_n(\hat{f}, \hat{r})}_{=:\Delta_f} + \widetilde{\mathcal{L}}_n(\hat{f}, \hat{r}) + \mathcal{O}_p(1/n). \tag{6}$$

We can handle $\Delta_r$ by the estimation error of $r^*$ as shown in Lemma 6 in Appendix B.4. We evaluate the projected MSE by combining (6) with Lemma 1 and Lemma 6:

**Theorem 1.** *Assume that the conditions of Lemmas 1–2 hold. Then, for any any $\delta \in (0, 1)$, there exists a constant $c > 0$ such that the following inequality holds with $n \geq 1$ and with probability at least $1 - \delta$, for any $\gamma \in (0, 1)$:*

$$\mathcal{L}(\hat{f}, r^*) \leq c \frac{\mathcal{C}_B(\mathcal{F}) + \mathcal{C}_B(\mathcal{R})}{\sqrt{n}} + \mathcal{O}_p\left(\max\left\{\sqrt{\frac{\log(1/\delta)}{n}}, \frac{1}{n^{1/(2+\gamma)}}\right\}\right).$$

This result reveals the following two findings: (i) the projected MSE is affected separately by the complexity of $\mathcal{F}$ and $\mathcal{R}$, and (ii) the overall convergence is $\mathcal{O}(1/\sqrt{n})$ when these complexities are finite. For instance, when using neural networks with Definition 1, the assumptions on the hypothesis classes are satisfied (Lemma 3).

## 5.3 FROM PROJECTED MSE TO MSE OF $f^*$

Following Chen & Pouzo (2012) and Dikkala et al. (2020, Appendix C.4), we discuss the derivation of the upper bound of the MSE of $f^*$ from the projected MSE. Let us define the measure of ill-posedness (Chen & Pouzo, 2012; Dikkala et al., 2020) with respect to the function class $\mathcal{F}$ as $\tau := \sup_{f \in \mathcal{R}} \frac{\|f - f^*\|_{L^2(O)}}{L(f, r^*)}$. Then, the MSE of $f^*$ is upper bounded as $\|f - f^*\|_{L^2(O)}^2 \leq \tau^2 L(\hat{f}, r^*)$ (Chen & Pouzo, 2012; Dikkala et al., 2020). Thus, Theorem 1 also implies

$$\|\hat{f} - f^*\|_{L^2(O)}^2 = c\tau^2 \frac{\mathcal{C}_B(\mathcal{F}) + \mathcal{C}_B(\mathcal{R})}{\sqrt{n}} + \mathcal{O}_p\left(\tau^2 \max\left\{\sqrt{\frac{\log(1/\delta)}{n}}, \frac{1}{n^{1/(2+\gamma)}}\right\}\right).$$

Note that Dikkala et al. (2020) derived $1/n$ rate on the MSE of $f^*$.

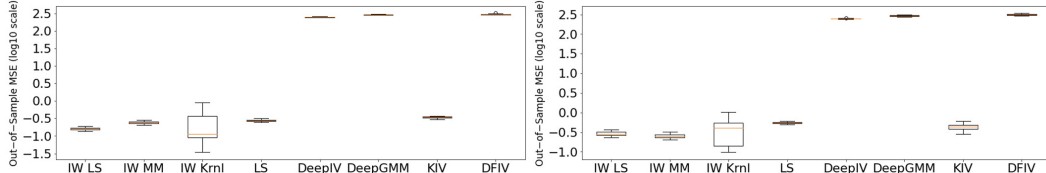

Figure 1: The log10 scaled MSEs of the setting in Newey & Powell (2003). The left graph shows the results using the original dataset. The right graph shows the results with additional IVs.

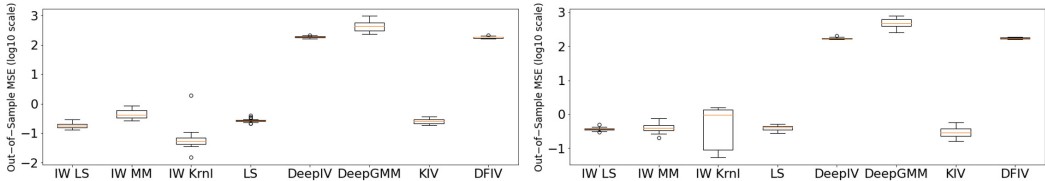

Figure 2: The log10 scaled MSEs of the original setting in Ai & Chen (2003). The left graph shows the result with $R = 0.1$ and the right graph shows the result with $R = 0.9$.

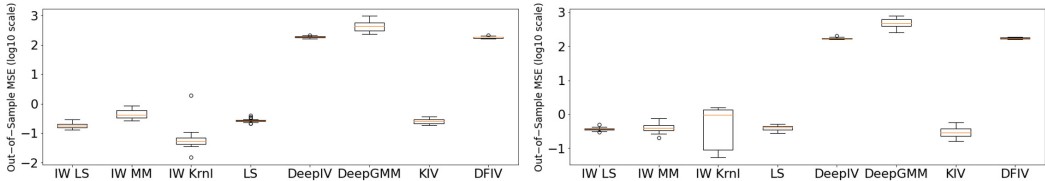

Figure 3: The log10 scaled MSEs of the setting in Ai & Chen (2003) with additional IVs. The left graph shows the result with $R = 0.1$ and the right graph shows the result with $R = 0.9$.

## 6 EXPERIMENTS

We implement the following three methods based on our proposed method: first, we use neural networks to predict $f^*$ and train the model by penalized least-squares in (4) (IW-LS); second, we use neural networks to predict $f^*$ and train the model by minimizing the sum of approximated moment restrictions in (2) (IW-MM), which is the same as IW-LS except for the penalized term in the IW-LS; third, we use a linear-in-parameter model with the Gaussian kernel to predict $f^*$ and train the model by GMM (IW-Krnl). For all cases, we use neural networks for estimating $r^*$. We compare our proposed methods with four methods: DeepGMM (Bennett et al. (2019)), DFIV (Xu et al. (2021a)), DeepIV (Hartford et al. (2017)), and KIV (Singh et al. (2019)). We use the datasets proposed in Newey & Powell (2003), Ai & Chen (2003), and Hartford et al. (2017). In addition, we train neural networks by simple least squares (LS), ignoring the dependency between $X_i$ and $\varepsilon_i$, as a comparison. To fairly evaluate the performances, for DeepGMM, DFIV, DeepIV, and KIV, we use the code and hyperparameters used in Xu et al. (2021a)[1]. For IW-LS, IW-MM, IW-Krnl, and LS, we also follow the model and hyperparameters of the code as possible. More details are shown in Appendix C.

### 6.1 EXPERIMENTS WITH DATASETS OF NEWEY & POWELL (2003) AND AI & CHEN (2003)

First, we investigate the performances of the proposed methods using econometric settings of Newey & Powell (2003) and Ai & Chen (2003). These settings have simpler structures than recently proposed settings, such as in Hartford et al. (2017). When using complex and high-dimensional datasets, there is an inherent difficulty in learning due to its complexity, separate from the problem setting. Therefore, we use simple datasets to check whether the proposed method can actually learn $f^*$.

Newey & Powell (2003) generates $\{(Y_i, X_i, Z_i)\}_{i=1}^n$ as follows: first, they generate $\{(\varepsilon_i, U_i, Z_i)\}_{i=1}^n$ from the multivariate normal distribution $\mathcal{N}\left(\begin{pmatrix} 0 \\ 0 \\ 0 \end{pmatrix}, \begin{pmatrix} 1 & 0.5 & 0 \\ 0.5 & 1 & 0 \\ 0 & 0 & 1 \end{pmatrix}\right)$; then, they generate $X_i = Z_i + U_i$ and $Y_i = f^*(X_i) + \varepsilon_i$, where $f^*(X_i) = \ln(|X_i - 1| + 1)\operatorname{sgn}(X_i - 1)$.

---

[1] https://github.com/liyuan9988/DeepFeatureIV

Figure 4: The log10 scaled MSEs of the demand design experiments with $1,000$ samples. The left graph show the results with $\rho = 0.25$ and the right graph shows the results with $\rho = 0.75$.

Ai & Chen (2003) generates $\{(Y_i, X_i, Z_i)\}_{i=1}^n$ as follows: first, they generate $\{(\varepsilon_i, X_{1i}, V_i, U_i)\}_{i=1}^n$ as $\varepsilon_i \sim \mathcal{N}\left(0, X_{1i}^2 + V_i^2\right)$, $X_{1i} \overset{\text{i.i.d.}}{\sim} \text{Unif}[0, 1]$, $V_i \overset{\text{i.i.d.}}{\sim} \text{Unif}[0, 1]$, and $U_i \overset{\text{i.i.d.}}{\sim} \mathcal{N}\left(0, X_{1i}^2 + V_i^2\right)$; second, they generate $X_{2i} = X_{1i} + V_i + R \times \varepsilon_i + U_i$ and $Y_1 = X_{1i}\gamma_0 + h_0\left(X_{2i}\right) + \varepsilon_i$, where $h_0\left(X_{2i}\right) = \exp\left(X_{2i}\right) / \left(1 + \exp\left(X_{2i}\right)\right)$ and $R$ is chosen as 0.9; then, obtain $X_i = \left(X_{1i} \; X_{2i}\right)^\top$ and $Z_i = \left(X_{1i} \; V_i\right)$. Here, $\varepsilon_i$ and $U_i$ are unobservable, and $f^*(X_i) = X_{1i}\gamma_0 + h_0\left(X_{2i}\right)$, where the function $h_0$ and $\gamma_0$ are unknown.

We run each algorithm 20 times on each dataset with $n = 1,000$ and calculate the mean squared error (MSE). In the left graph of Figure 1 and Figure 2, we report the MSEs. In the dataset of Newey & Powell (2003), DFIV and our proposed IW-MM produce smaller MSE. The dataset only has a one-dimensional $X_i$, which mitigates the identification problem of neural networks. In contrast, in the dataset of Ai & Chen (2003), IW-LS leads to smaller MSE. In Appendix C.4, using the experimental setting of Newey & Powell (2003), we also show additional experimental results on the empirical convergence of the MSE of the IWMM and a comparison between the IWMM and classical 2SLS.

Because the dimensions of the IVs are low in the original settings, we add more IVs to the original settings and investigate the performances of the algorithms. The detailed settings are described in Appendix C.1. The results are shown in the right graph of Figures 1 and Figure 3.

## 6.2 SIMULATION STUDIES USING DEMAND DESIGN DATASETS

To investigate the performance with more complicated and high-dimensional datasets, we use the demand design dataset for synthetic airline ticket sales proposed by Hartford et al. (2017). In this dataset, we observe $(Y_i, P_i, T_i, S_i, C_i)$, where $Y_i$ is sales, $P_i$ is price, $T_i$ is time, $S_i$ is consumer's emotion, and $C_i$ is cost to use as IV. Here, $T_i$ and $S_i$ are covariates, that is, $X_i = (T_i, S_i)$, and the IV is $Z_i = (C_i, T_i, S_i)$. The sales $Y_i$ is generated as $Y_i = 100 + (10 + P_i)S_i h(T_i) - 2P_i + \varepsilon_i$, where $h(t) = 2\left(\frac{(t-5)^4}{600} + \exp\left(-4(t-5)^2\right) + \frac{t}{10} - 2\right)$. Since $P_i$ is an endogenous variable and correlated with the IV, we generate $P_i$ to contain $C_i$ as $P_i = 25 + (C_i + 3)h(T_i) + V_i$. In the simulation, we assume $\varepsilon_i \sim \mathcal{N}\left(\rho V_i, 1 - \rho^2\right)$, $V_i \sim \mathcal{N}(0, 1)$, $C_i \sim \mathcal{N}(0, 1)$, $T_i$ is sampled from the uniform distribution with the continuous support $[0, 10]$, and $S_i$ is sampled from the uniform distribution with the discrete support $\{1, \ldots, 7\}$. Here note that $\mathbb{E}[\varepsilon_i | C_i, T_i, S_i] = \mathbb{E}[\varepsilon_i | Z_i] = 0$. The extent of correlation between $P_i$ and $V_i$ is controlled by $\rho \in \{0.25, 0.75\}$ and the larger the $\rho$ is, the more severe the correlation problem becomes.

Figure 4 shows the results with $1,000$ samples. The results under other settings are reporetd in Appendix C.2. In this setting, IW-LS and LS outperform the other methods. We consider this is because $f^*(X_i)$ takes large values compared to the error term. Under this situation, a model trained to predict $Y_i$ may perform well because the influence of $\mathbb{E}[\varepsilon_i | X_i] \neq 0$ is limited. However, the purpose of using NPIV in the first place is because the influence is large, or else effects of $\mathbb{E}[\varepsilon_i | X_i] \neq 0$ can be ignored. Therefore, this dataset is often used in the existing studies, it may not be suitable.

## 7 CONCLUSION

This paper proposed a method for learning causal relationships from conditional moment restrictions. Our method is based on importance weighting using the conditional density ratio. The proposed method showed superior performance in experiments. We point out potential problems in recently proposed methods concerning identification and empirical performances.

ACKNOWLEDGMENTS

The authors would like to thank Liyuan Xu for his constructive advice.

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

## A    LITERATURE REVIEW OF IV METHODS

In the field of economics, the causal effect is often referred to as the structural effect, because the causal relationship arises from some economic structure. The idea of endogeneity has a close relationship with the estimation of the structural effect. For example, in estimation problems with supply/demand models, because the supply and demand are determined simultaneously, there is a simultaneous equation bias, which causes the correlation between the explanatory variable and error term. This correlation is called enodogeneity. Under the enodogeneity, the OLS does not yield a consistent estimator of the structural model. To obtain a consistent estimator to capture the causal effect, the method of IVs have been used Wright (1928); Reiersöl (1945).

### A.1    IV METHODS FOR LINEAR STRUCTURAL MODELS

First, we consider a linear structural model. For two random variables $Y \in \mathbb{R}$, $X \in \mathbb{R}^d$, endogeneity refers to a situation, such that

$$Y = X^\top \beta + \varepsilon_i, \tag{7}$$

where $\varepsilon_i \in \mathbb{R}$ is the error term, $\beta$ is the parameter of interest, and

$$\mathbb{E}[X\varepsilon] \neq 0. \tag{8}$$

This regression model is called a structural equation and $\beta$ is the structural parameter. Under endogeneity, the least squares method does not yield a consistent estimator.

Let us consider an IV $Z \in \mathbb{R}^k$ such that

$$\mathbb{E}[Z\varepsilon] = 0. \tag{9}$$

For the estimation of the structural parameter under endogeneity, the IV $Z$ plays an important.

### A.2    IV METHODS FOR NONPARAMETRIC STRUCTURAL MODELS

Nonparametric estimation of structural models under conditional moment restrictions is an important topic in statistics and econometrics, because it allows us to model economic relationships flexibly. Let $h^*(\cdot) = (h_1^*(\cdot), \ldots, h_q^*(\cdot))$ be unknown nonparametric structural functions, where each function $h_\ell^*(\cdot)$ may depend on $X_i$ and $Y_i$. In general, we can consider estimating the unknown nonparametric structural functions $h^*(\cdot)$ defined as

$$\mathbb{E}\left[\rho(Y, X; \theta_0, h^*(\cdot))|Z\right] = 0, \tag{10}$$

where $\rho(\cdot; \theta_0, h^*(\cdot)))$ is a vector of residuals with functional forms. Note that the conditional distribution of $Y$ given $X$ is not specified.

This model is a generalization of semi/nonparametric estimation with conditional moment restrictions considered in Chamberlain (1992) and Newey & Powell (2003). This model includes many structural models as special cases. A typical example is the target structural model of the NPIV problem defined in Section 2 (Newey & Powell, 2003; Darolles et al., 2011). The NPIV problem also includes the estimation of a shape-invariant system of Engel curves with an endogenous total expenditure (Blundell et al., 2007). Another important special case of (10) is the quantile instrumental variables (IV) treatment effect model of Chernozhukov & Hansen (2005), and the nonparametric quantile instrumental variables regression of Chernozhukov et al. (2007) and Horowitz & Lee (2007a). There are other applications, such as asset pricing models (Chen & Ludvigson2009Ludvigson, 2009) and reinforcement learning.

There are three approaches to the estimation of the structural models under the conditional moment restrictions: the sieve minimum distance (SMD) method, the function space Tikhonov regularized minimum distance (TMD) method, and the minimax optimization method.

**SMD method.**    The SMD procedure minimizes a consistent estimate of the minimum distance criterion over some finite-dimensional compact sieve space. Newey & Powell (2003) extends the 2SLS methods for linear structural models to the NPIV problem. The authors approximate the nonparametric structural function with a sieve estimator, which is a linear combination of growing

feature basis functions. Ai & Chen (2003) and Chen & Pouzo (2012) propose the penalized minimum distance sieve estimator to a more general problem defined as (10).

When applying the minimum distance method to the NPIV problem, we consider solving the following problem:
$$\min_{h \in \mathcal{H}} \mathbb{E}_Z[\mathbb{E}_{Y,X}[Y - h(X)|Z]^2] + \lambda R(h),$$

where $R(h)$ is a regularizer. Chen & Pouzo (2012) approximates the function class $\mathcal{H}_n$ by linear functions in growing feature space with the sample size $n$. Subsequently, the authors also estimate the function $m(z) = \mathbb{E}[y - h(x) \mid z]$ based on another growing sieve.

**TMD method.** In the TMD method, we minimize a consistent penalized estimate of the minimum distance criterion over the whole infinite dimensional function space $\mathcal{H}$, in which the penalty function is of the classical Tikhonov type (Hall & Horowitz, 2005; Horowitz & Lee, 2007b; Carrasco et al., 2007b; Darolles et al., 2011; Gagliardini & Scaillet, 2012). In the NPIV problem, this approach is equivalent to solving
$$\min_{h \in \mathcal{H}} \mathbb{E}_Z[\mathbb{E}_{Y,X}[Y - h(X)|Z]] + \lambda\|h\|_2^2. \tag{11}$$

Note that this is equivalent to the minimum distance estimation, such as Chen & Pouzo (2012).

**Minimax optimization method.** Recently, in the machine learning literature, the minimax approach has garnered attention (Bennett et al., 2019; Dikkala et al., 2020). In this approach, we solve the following minimax optimization problem:
$$\min_{f \in \mathcal{F}} \max_{h \in \mathcal{H}} \mathbb{E}[(Y - f(X))h(Z)] - \|h\|_2^2. \tag{12}$$

Such a representation of the conditional moment restrictions has long been used in statistics and econometrics. One of the related studies is the specification testing by Bierens (1982; 1990). The authors assume that a statistical model that satisfies the conditional moment restrictions is correct and propose a specification testing method to investigate whether the model is correct. These studies transform the conditional moment restrictions to unconditional moment restrictions by using the product of any function of a random variable used in the conditioning and the moment function. This idea of transformation to unconditional moment restrictions is also used in other related work, such as Santos (2012).

In particular, Dikkala et al. (2020) shows the minimax optimality of a method based on such minimax optimization. This minimax optimization technique is known to be applicable, not only to the NPIV problem, but also to a wider range of problems. It is also closely related to debiased machine learning literature (Chernozhukov et al., 2020).

## B   PROOFS IN SECTION 5

### B.1   MATHEMATICAL TOOLS

**Concentration inequalities for empirical processes**   Given a probability distribution $P$ and a random variable $g(X)$, we denote the expectation of $g(X)$ under $P$ by $\int g \mathrm{d}P$. Given samples $X_1, X_2, \ldots, X_n$ from $P$, the empirical distribution is denoted by $P_n$, and the empirical mean is denoted by $\int g \mathrm{d}P_n$; that is, $\int g \mathrm{d}P_n = \frac{1}{n}\sum_{i=1}^n g(X_i)$. We also denote $\int g \mathrm{d}P - \frac{1}{n}\sum_{i=1}^n g(X_i)$ by $\int g \mathrm{d}(P - P_n)$.

We also define
$$\rho_K^2(g) = 2K^2 \int \left(\exp\left(|g|/K - 1 - |g|/K\right)\right) \mathrm{d}P, \qquad K > 0. \tag{13}$$

Let $\mathcal{G}$ satisfy
$$\sup_{g \in \mathcal{G}} \rho_K(g) \leq R \tag{14}$$

Then, we summarize some tools used in the theoretical analysis.

**Proposition 1** (Theorem 5.11 in van de Geer (2000)). *$C$ is a (sufficiently large) universal constant, whereas $a$, $C_0$, and $C_1$ may be chosen, but do have to satisfy the following conditions:*

- $a \leq C_1 \sqrt{n} R^2 / K$;

- $a \leq 8 \sqrt{n} R$;

- $a \geq C_0 \left( \max \left\{ \int_{a/(2^6 \sqrt{n})}^{R} \sqrt{\log \mathcal{N}(u, \mathcal{G}, \| \cdot \|)} \mathrm{d}u, R \right\} \right)$

- $C_0^2 \geq C^2(C_1 + 1)$.

*Then,*

$$\mathbb{P} \left( \sup_{g \in \mathcal{G}} \left| \sqrt{n} \int g \mathrm{d}(P_n - P) \right| \geq a \right) \leq C \exp \left( -\frac{a^2}{C^2(C_1 + 1)R^2} \right) \tag{15}$$

**Proposition 2** (Lemma 5.14 in van de Geer (2000)). *Let $\mathcal{G} \subset L^2(P)$ be a function class and the map $I(g)$ be a complexity measure of $g \in \mathcal{G}$, where $I$ is a non-negative function on $\mathcal{R}$ and $I(g_0) < \infty$ for a fixed $g_0 \in \mathcal{G}$. We now define $\mathcal{G}_M = \{g \in \mathcal{G} : I(g) \leq M\}$ satisfying $\mathcal{G} = \bigcup_{M \geq 1} \mathcal{G}_M$. Suppose that there exist $c_0 > 0$ and $0 < \gamma < 2$ such that*

$$\sup_{g \in \mathcal{G}_M} \|g - g_0\| \leq c_0 M, \qquad \sup_{\substack{r \in \mathcal{G}_M \\ \|g - g_0\|_{L^2(P)} \leq \delta}} \|g - g_0\|_\infty \leq c_0 M, \quad \text{for all } \delta > 0,$$

*and that $\log \mathcal{N}(\delta, \mathcal{G}_M, P) = \mathcal{O}\left( M/\delta \right)^\gamma$. Then, we have*

$$\sup_{g \in \mathcal{G}} \frac{\left| \int (g - g_0) \mathrm{d}(P - P_n) \right|}{D(g)} = \mathcal{O}_p(1), \quad (n \to \infty),$$

*where $D(g)$ is defined by*

$$D(g) = \max \left\{ \frac{\|g - g_0\|_{L^2(P)}^{1 - \gamma/2} I(g)^{\gamma/2}}{\sqrt{n}}, \frac{I(g)}{n^{2/(2+\gamma)}} \right\}.$$

**Lemma 3** (Lemma 9 of Kato & Teshima (2021)). *Let $\ell : (b_r, B_r) \to \mathbb{R}$ be a $\nu$-Lipschitz continuous function. Let $\log \mathcal{N}\left( \delta, \mathcal{F}, \| \cdot \|_{L^2(P)} \right)$ denote the bracketing entropy of $\mathcal{F}$ with respect to a distribution $P$. Then, for any distribution $P$, any $\gamma > 0$, any $M \geq 1$, and any $\delta > 0$, we have*

$$\log \mathcal{N}\left( \delta, \ell \circ \mathcal{H}_M, \| \cdot \|_{L^2(P)} \right) \leq \frac{(s+1)(2\nu)^\gamma}{\gamma} \left( \frac{M}{\delta} \right)^\gamma.$$

*Moreover, there exists $c_0 > 0$ such that for any $M \geq 1$ and any distribution $P$,*

$$\sup_{\ell \circ r \in \ell \circ \mathcal{H}_M} \|\ell \circ r - \ell \circ r^*\|_{L^2(P)} \leq c_0 \nu M,$$

$$\sup_{\substack{\ell \circ r \in \ell \circ \mathcal{H}_M \\ \|\ell \circ r - \ell \circ r^*\|_{L^2(P)} \leq \delta}} \|\ell \circ r - \ell \circ r^*\|_\infty \leq c_0 \nu M, \quad \text{for all } \delta > 0.$$

**Concentration inequalities for U-statistics.** Given a probability distributions $P$ and $Q$ and a random variable $g(X, Z)$, we denote the expectation of $g(X, Z)$ under $P$ and $Q$ by $\int \int g \mathrm{d}P \mathrm{d}Q$; that is, $\mathbb{E}_X \left[ \mathbb{E}_Z \left[ g(X, Z) \right] \right] = \int \int g \mathrm{d}P \mathrm{d}Q$. Given samples $X_1, X_2, \ldots, X_n$ from $P$ and $Z_1, Z_2, \ldots, Z_n$ from $Q$, the empirical distributions are denoted by $P_n$ and $Q_n$, and the empirical mean is denoted by $\int \int g \mathrm{d}P_n \mathrm{d}Q_n$; that is, $\int \int g \mathrm{d}P_n \mathrm{d}Q_n = \frac{1}{n} \sum_{j=1}^n \frac{1}{n} \sum_{i=1}^n g(X_i, Z_j)$. We also denote $\int \int g \mathrm{d}P \mathrm{d}Q - \frac{1}{n} \sum_{j=1}^n \frac{1}{n} \sum_{i=1}^n g(X_i, Z_j)$ by $\int \int g \mathrm{d}(P - P_n) \mathrm{d}(Q - Q_n)$.

We also define

$$\rho_K^2(g) = 2K^2 \int \left( \exp \left( |g|/K - 1 - |g|/K \right) \right) \mathrm{d}P, \qquad K > 0. \tag{16}$$

Let $\mathcal{G}$ satisfy

$$\sup_{g \in \mathcal{G}} \rho_K(g) \leq R \tag{17}$$

Then, we summarize some tools used in the theoretical analysis.

**Proposition 3** (Concentration inequality on empirical processes with U-statistics: Theorem 5 in Arcones (1995), adjusted to our setting). *Suppose that $S_1, ..., S_n$ are $\mathcal{S}$-valued i.i.d. random variables and consider a class of symmetric functions $\mathcal{H} \subset L^2(\mathcal{S} \times \mathcal{S})$. Also, suppose that any function in $\mathcal{H}$ is uniformly bounded by $b > 0$ and define $\sigma^2 = \sup_{h \in \mathcal{H}} \mathrm{Var}_S(\mathbb{E}_{S'}[h(S, S')])$ where $S'$ is an i.i.d. copy of $S$. If the $\mathcal{N}(\varepsilon, \mathcal{H}, \|\cdot\|_2) \leq (A/\varepsilon)^\nu$ for some $A, \nu > 0$, for any $t \geq c$ with some $c > 0$, we obtain*

$$\mathbb{P}\left(n^{1/2} \sup_{h \in \mathcal{H}} \left\{ \frac{1}{n(n-1)} \sum_{i,i'=1, i \neq i'}^n h(S_i, S_{i'}) - \mathbb{E}_{S,S'}[h(S,S)] \right\} \geq t \right)$$

$$\leq 8 \exp(-t^2/2^{17}(\sigma^2 + tbn^{-1/2})) + 8A^{2\nu}(\sigma^2 + 2tbn^{-1/2})^{-\nu} \exp(-n(\sigma^2 + tbn^{-1/2}/2)/2b^2)$$

$$+ 2\exp(-t^2/(2^{11}bc(\sigma^2 + tbn^{-1/2}))).$$

**Proposition 4** (Bernsteins' inequality for U-statistics (Hoeffding, 1963; Arcones & Giné, 1993)). *Let $\|g\|_\infty < c$, $\int \int g \mathrm{d}P \mathrm{d}Q = 0$, and $\sigma^2 = \int \int g^2 \mathrm{d}P \mathrm{d}Q$. Then, for any $a' > 0$:*

$$\mathbb{P}\left( \int \int g \mathrm{d}P_n \mathrm{d}Q_n > a' \right) \leq \exp\left( \frac{na'^2/2}{2\sigma^2 + (2/3)ca'} \right).$$

This implies that for any $a > 0$,

$$\mathbb{P}\left( \left| \int \int g - g_0 \mathrm{d}\left(P - P_n\right) \mathrm{d}\left(Q - Q_n\right) \right| > \frac{a}{\sqrt{n}} \right) \leq \exp\left( \frac{a^2/2}{6\sigma^2 + 2ca/\sqrt{n}} \right).$$

As well as Suzuki et al. (2008), by applying this result, we can obtain the following result.

**Proposition 5** (From Proof of Theorem 1 in Suzuki et al. (2008)). *Let $\mathcal{G} \subset L^2(P \times Q)$ be a function class and the map $I(g)$ be a complexity measure of $g \in \mathcal{G}$, where $I$ is a non-negative function on $\mathcal{R}$ and $I(g_0) < \infty$ for a fixed $g_0 \in \mathcal{G}$. We now define $\mathcal{G}_M = \{g \in \mathcal{G} : I(g) \leq M\}$ satisfying $\mathcal{G} = \bigcup_{M \geq 1} \mathcal{G}_M$. Suppose that there exist $c_0 > 0$ and $0 < \gamma < 2$ such that*

$$\sup_{g \in \mathcal{G}_M} \|g - g_0\| \leq c_0 M, \qquad \sup_{\substack{r \in \mathcal{G}_M \\ \|g - g_0\|_{L^2(P \times Q)} \leq \delta}} \|g - g_0\|_\infty \leq c_0 M, \quad \text{for all } \delta > 0,$$

*and that $\log \mathcal{N}(\delta, \mathcal{G}_M, P \times Q) = \mathcal{O}\left( M/\delta \right)^\gamma$. Then, we have*

$$\sup_{g \in \mathcal{G}} \frac{\left| \int \int (g - g_0) \mathrm{d}(P - P_n) \mathrm{d}(Q - Q_n) \right|}{D(g)} = \mathcal{O}_p(1), \ (n \to \infty),$$

*where $D(g)$ is defined by*

$$D(g) = \max\left\{ \frac{\|g - g_0\|_{L^2(P \times Q)}^{1-\gamma/2} I(g)^{\gamma/2}}{\sqrt{n}}, \frac{I(g)}{n^{2/(2+\gamma)}} \right\}.$$

$$\sup_{g \in \mathcal{R}} \left| \int \int (g - g_0) \mathrm{d}(P - P_n) \mathrm{d}(Q - Q_n) \right| = \mathcal{O}_p\left( 1/\sqrt{n} \right), \ (n \to \infty). \tag{18}$$

**Complexities of neural networks.**

**Definition 1** (ReLU neural networks; Schmidt-Hieber, 2020). *For $L \in \mathbb{N}$ and $p = (p_0, \ldots, p_{L+1}) \in \mathbb{N}^{L+2}$,*

$$\mathcal{F}(L, p) := \{f : x \mapsto W_L \sigma_{v_L} W_{L-1} \sigma_{v_{L-1}} \cdots W_1 \sigma_{v_1} W_0 x :$$

$$W_i \in \mathbb{R}^{p_{i+1} \times p_i}, v_i \in \mathbb{R}^{p_i}(i = 0, \ldots, L)\},$$

*where $\sigma_v(y) := \sigma(y - v)$, and $\sigma(\cdot) = \max\{\cdot, 0\}$ is applied in an element-wise manner. Then, for $s \in \mathbb{N}, F \geq 0, L \in \mathbb{N}$, and $p \in \mathbb{N}^{L+2}$, define*

$$\mathcal{H}(L, p, s, F) := \{f \in \mathcal{F}(L, p) : \sum_{j=0}^L \|W_j\|_0 + \|v_j\|_0 \leq s, \|f\|_\infty \leq F\},$$

where $\| \cdot \|_0$ denotes the number of non-zero entries of the matrix or the vector, and $\| \cdot \|_\infty$ denotes the supremum norm. Now, fixing $\bar{L}, \bar{p}, s \in \mathbb{N}$ as well as $F > 0$, we define

$$\mathrm{Ind}_{\bar{L},\bar{p}} := \{(L,p) : L \in \mathbb{N}, L \leq \bar{L}, p \in [\bar{p}]^{L+2}\},$$

and we consider the hypothesis class

$$\bar{\mathcal{H}} := \bigcup_{(L,p) \in \mathrm{Ind}_{\bar{L},\bar{p}}} \mathcal{H}(L,p,s,F)$$

$$\mathcal{H} := \{r \in \bar{\mathcal{H}} : \mathrm{Im}(r) \subset (b_r, B_r)\}.$$

Moreover, we define $I_1 : \mathrm{Ind}_{\bar{L},\bar{p}} \to \mathbb{R}$ and $I : \mathcal{H} \to [0, \infty)$ by

$$I_1(L,p) := 2|\mathrm{Ind}_{\bar{L},\bar{p}}|^{\frac{1}{s+1}}(L+1)V^2,$$

$$I(r) := \max \left\{ \|r\|_\infty, \min_{\substack{(L,p) \in \mathrm{Ind}_{\bar{L},\bar{p}} \\ r \in \mathcal{H}(L,p,s,F)}} I_1(L,p) \right\},$$

where $V := \prod_{l=0}^{L+1}(p_l + 1)$, and we define

$$\mathcal{H}_M := \{r \in \mathcal{H} : I(r) \leq M\}.$$

**Lemma 4** (Lemma 5 in Schmidt-Hieber (2020)). *For $L \in \mathbb{N}$ and $p \in \mathbb{N}^{L+2}$, let $V := \prod_{l=0}^{L+1}(p_l + 1)$. Then, for any $\delta > 0$,*

$$\log \mathcal{N}(\delta, \mathcal{H}(L,p,s,\infty), \| \cdot \|_\infty) \leq (s+1)\log(2\delta^{-1}(L+1)V^2).$$

**Definition 2** (Derived function class and bracketing entropy). *Given a real-valued function class $\mathcal{F}$, define $\ell \circ \mathcal{F} := \{\ell \circ f : f \in \mathcal{F}\}$. By extension, we define $I : \ell \circ \mathcal{H} \to [1, \infty)$ by $I(\ell \circ r) = I(r)$ and $\ell \circ \mathcal{H}_M := \{\ell \circ r : r \in \mathcal{H}_M\}$. Note that, as a result, $\ell \circ \mathcal{H}_M$ coincides with $\{\ell \circ r \in \ell \circ \mathcal{H} : I(\ell \circ r) \leq M\}$.*

**Notations.** Let us denote a pair of random variables $(Y_i, X_i)$ by $W_i$. We denote the distribution of $(W_i, Z_i)$ by $R$ and its empirical distribution as $O_n$. Besides, we denote the distributions of $W_i$ and $Z_i$ by $P$ and $Q$, and their empirical distributions by $P_n$ and $Q_n$. For a function $g(X, Z)$, we define the $L^2$ risk over the distribution $P$ and $Q$ as

$$\|g\|_{L^2(P \times Q)} = \sqrt{\mathbb{E}_X\left[\mathbb{E}_Z\left[g^2(X,Z)\right]\right]} = \sqrt{\int \int g^2 \mathrm{d}P \mathrm{d}Q}.$$

## B.2 PROOF OF LEMMA 1: ESTIMATION ERROR IN CONDITIONAL DENSITY RATIO ESTIMATION

**Lemma 5** (Decomposition of MSE).

$$\|\hat{r} - r^*\|_{L^2(O)}^2 \leq \left| \mathbb{E}_{W,Z}\left[r^*(W|Z) - \hat{r}(W|Z)\right] - \frac{1}{n}\sum_{i=1}^n \left(r^*(W_i|Z_i) - \hat{r}(W_i|Z_i)\right) \right|$$

$$+ \frac{1}{2}\left| \mathbb{E}_Z\left[\mathbb{E}_W\left[r^{*2}(W|Z) - \hat{r}^2(W|Z)\right]\right] - \frac{1}{n}\sum_{j=1}^n \frac{1}{n}\sum_{i=1}^n \left(r^{*2}(W_i|Z_j) - \hat{r}^2(W_i|Z_j)\right) \right|.$$

*Proof of Lemma 5.* Since The estimator $\hat{r}$ is the minimizer of the empirical risk, it satisfies the inequality

$$-\frac{1}{n}\sum_{i=1}^n \hat{r}(W_i|Z_i) + \frac{1}{2}\frac{1}{n}\sum_{j=1}^n \frac{1}{n}\sum_{i=1}^n \hat{r}^2(W_i|Z_j) \leq -\frac{1}{n}\sum_{i=1}^n r^*(W_i|Z_i) + \frac{1}{2}\frac{1}{n}\sum_{j=1}^n \frac{1}{n}\sum_{i=1}^n r^{*2}(W_i|Z_j).$$

(19)

Additionally, we consider an expectation of squared residuals as

$$
\frac{1}{2}\mathbb{E}_Z\left[\mathbb{E}_W\left[\left(\hat{r}(W|Z) - r^*(W|Z)\right)^2\right]\right]
$$

$$
= \frac{1}{2}\mathbb{E}_Z\left[\mathbb{E}_W\left[\hat{r}^2(W|Z) - 2\hat{r}(W|Z)r^*(W|Z) + r^{*2}(W|Z)\right]\right]
$$

$$
= \frac{1}{2}\mathbb{E}_Z\left[\mathbb{E}_W\left[\hat{r}^2(W|Z)\right]\right] - \mathbb{E}_{W,Z}\left[\hat{r}(W|Z)\right] + \frac{1}{2}\mathbb{E}_{W,Z}\left[r^*(W|Z)\right]
$$

$$
\quad + \mathbb{E}_{W,Z}\left[r^*(W|Z)\right] - \mathbb{E}_{W,Z}\left[r^*(W|Z)\right]
$$

$$
= \frac{1}{2}\mathbb{E}_Z\left[\mathbb{E}_W\left[\hat{r}^2(W|Z)\right]\right] - \mathbb{E}_{W,Z}\left[\hat{r}(W|Z)\right] - \frac{1}{2}\underbrace{\mathbb{E}_{W,Z}\left[r^*(W|Z)\right]}_{=\mathbb{E}_Z[\mathbb{E}_W[r^{*2}(W|Z)]]} + \mathbb{E}_{W,Z}\left[r^*(W|Z)\right]
$$

$$
= -\frac{1}{2}\mathbb{E}_Z\left[\mathbb{E}_W\left[r^{*2}(W|Z) - \hat{r}^2(W|Z)\right]\right] + \mathbb{E}_{W,Z}\left[r^*(W|Z) - \hat{r}(W|Z)\right]. \tag{20}
$$

Taking sum of the both hand sides of (19) and (20), then we obtain the following by subtracting $-\frac{1}{n}\sum_{i=1}^n \hat{r}(W_i|Z_i) + \frac{1}{2}\frac{1}{n}\sum_{j=1}^n\frac{1}{n}\sum_{i=1}^n \hat{r}^2(W_i|Z_j)$ from the both side as

$$
\frac{1}{2}\mathbb{E}_{W,Z}\left[\left(\hat{r}(W|Z) - r^*(W|Z)\right)^2\right]
$$

$$
\leq -\frac{1}{2}\mathbb{E}_Z\left[\mathbb{E}_W\left[r^{*2}(W|Z) - \hat{r}^2(W|Z)\right]\right] + \mathbb{E}_{W,Z}\left[r^*(W|Z) - \hat{r}(W|Z)\right]
$$

$$
\quad + \frac{1}{2}\frac{1}{n}\sum_{j=1}^n\frac{1}{n}\sum_{i=1}^n\left(r^{*2}(W_i|Z_j) - \hat{r}^2(W_i|Z_j)\right) - \frac{1}{n}\sum_{i=1}^n\left(r^*(W_i|Z_i) - \hat{r}(W_i|Z_i)\right).
$$

Then, we obtain the statement. $\qquad\square$

Our remaining task is to bound the following target values:

$$
\sup_{r\in\mathcal{R}}\left|\mathbb{E}_Z\left[\mathbb{E}_W\left[r^2(W|Z)\right]\right] - \frac{1}{n}\sum_{j=1}^n\frac{1}{n}\sum_{i=1}^n r^2(W_i|Z_j)\right|, \text{ and} \tag{21}
$$

$$
\sup_{r\in\mathcal{R}}\left|\mathbb{E}_{W,Z}\left[r(W|Z)\right] - \frac{1}{n}\sum_{i=1}^n r(W_i|Z_i)\right|. \tag{22}
$$

*Proof of Lemma 1.* Since $0 < \gamma < 2$, we can apply Propositions 2 and 5 in combination with Lemma 3 to obtain

$$
\sup_{r\in\mathcal{H}}\frac{\left|\mathbb{E}_Z\left[\mathbb{E}_W\left[r^2(W|Z)\right]\right] - \frac{1}{n}\sum_{j=1}^n\frac{1}{n}\sum_{i=1}^n r^2(W_i|Z_j)\right|}{D_1(r)} = \mathcal{O}_p(1),
$$

$$
\sup_{r\in\mathcal{H}}\frac{\left|\mathbb{E}_{W,Z}\left[r(W|Z)\right] - \frac{1}{n}\sum_{i=1}^n r(W_i|Z_i)\right|}{D_2(r)} = \mathcal{O}_p(1),
$$

where

$$
D_1(r) = \max\left\{\frac{\|r^2 - r^{*2}\|_{L^2(P\times Q)}^{1-\gamma/2}I(r^2)^{\gamma/2}}{\sqrt{n}}, \frac{I(r^2)}{n^{2/(2+\gamma)}}\right\}, \text{ and}
$$

$$
D_2(r) = \max\left\{\frac{\|r - r^*\|_{L^2(O)}^{1-\gamma/2}I(r)^{\gamma/2}}{\sqrt{n}}, \frac{I(r)}{n^{2/(2+\gamma)}}\right\}.
$$

Noting that $\sup_{r \in \mathcal{H}} I(r) < \infty$ and $\sup_{r \in \mathcal{H}} I(r^2) < \infty$. Then, for any $0 < \gamma < 2$, we have

$$\left\| \mathbb{E}_Z \left[ \mathbb{E}_W \left[ \hat{r}^2(W|Z) \right] \right] - \frac{1}{n} \sum_{j=1}^n \frac{1}{n} \sum_{i=1}^n \hat{r}^2(W_i|Z_j) \right\|_{L^2(P \times Q)}^2$$

$$\lesssim \mathcal{O}_p \left( \max \left\{ \frac{\|\hat{r}^2 - r^{*2}\|_{L^2(P \times Q)}^{1-\gamma/2}}{\sqrt{n}}, \frac{1}{n^{2/(2+\gamma)}} \right\} \right), \text{ and }$$

$$\left\| \mathbb{E}_{W,Z} \left[ \hat{r}(W|Z) \right] - \frac{1}{n} \sum_{i=1}^n \hat{r}(W_i|Z_i) \right\|_{L^2(P \times Q)}^2$$

$$\lesssim \mathcal{O}_p \left( \max \left\{ \frac{\|\hat{r} - r^*\|_{L^2(O)}^{1-\gamma/2}}{\sqrt{n}}, \frac{1}{n^{2/(2+\gamma)}} \right\} \right).$$

These inequalities imply

$$\|\hat{r} - r^*\|_{L^2(P \times Q)} \leq \mathcal{O}_p \left( n^{-\frac{1}{2+\gamma}} \right).$$

$\qquad\qquad\qquad\qquad\qquad\qquad\qquad\qquad\qquad\qquad\qquad\qquad\qquad\qquad\qquad\qquad$ □

## B.3 PROOF OF LEMMA 2

We start with the expected loss function for a learner $f : \mathcal{X} \to \mathbb{R}$ and a conditional density function $r : \mathcal{X} \times \mathcal{Y} \times \mathcal{Z} \to \mathbb{R}$ as

$$\mathcal{L}(f, r) := \mathbb{E}_Z \left[ \left( \mathbb{E}_W \left[ (f^*(X) - f(X)) r(W|Z) \right] \right)^2 \right], \tag{23}$$

where $W = (Y, X)$. For $T \in \mathbb{T}$, we consider $T$ i.i.d. random variables $W_i = (X_i, Y_i), i = 1, ..., n$ generated from a conditional distribution $P$ and approximate the inner expectation term as

$$\mathbb{E}_W \left[ (f^*(X) - f(X)) r(W|Z) \right] = n^{-1} \sum_{i=1}^n (f^*(X_i) - f(X_i)) r(W_i|Z) + \kappa_n(f, r),$$

where

$$\kappa_n(f, r) := \mathbb{E}_W \left[ (f^*(X) - f(X)) r(W|Z) \right] - n^{-1} \sum_{i=1}^n (f^*(X_i) - f(X_i)) r(W_i|Z)$$

is a residual. We substitute this form into (3) and obtain

$$\mathcal{L}(f, r) = \mathbb{E}_Z \left[ \left( n^{-1} \sum_{i=1}^n (f^*(X_i) - f(X_i)) r(W_i|Z) + \kappa_n(f, r) \right)^2 \right]$$

$$= \mathbb{E}_Z \left[ n^{-2} \sum_{k,k'=1}^T (f^*(X_i) - f(X_i)) r(W_i|Z)(f^*(X_{i'}) - f(X_{i'})) r(W_{i'}|Z) \right]$$

$$+ \underbrace{2\mathbb{E}_Z \left[ \kappa_n(f, r) n^{-1} \sum_{i=1}^n (f^*(X_i) - f(X_i)) r(W_i|Z) \right]}_{=:\mathcal{T}_1} + \underbrace{\mathbb{E}_Z[\kappa_n(f, r)^2]}_{=:\mathcal{T}_2}$$

$$= \underbrace{\mathbb{E}_Z \left[ n^{-2} \sum_{i,i'=1, i' \neq i}^n (f^*(X_i) - f(X_i)) r(W_i|Z)(f^*(X_{i'}) - f(X_{i'})) r(W_{i'}|Z) \right]}_{=:\mathcal{T}^*}$$

$$+ \underbrace{\mathbb{E}_Z \left[ n^{-2} \sum_{i=1}^n (f^*(X_i) - f(X_i))^2 r(W_i|Z)^2 \right]}_{=:\mathcal{T}_0} + \mathcal{T}_1 + \mathcal{T}_2.$$

For $\mathcal{T}_0$, the uniformly bounded property of $f^*$, $f$ and $r$ implies that it is $\mathcal{T}_0 = \mathcal{O}(1/n)$. For $\mathcal{T}_1$ and $\mathcal{T}_2$, the VC-class property and the Glivenko-Cantelli theorem (Section 2.8.1 in van der vaart & Wellner (1996)), we obtain $\kappa_n(f, r) \to 0$ in probability as $T \to \infty$ uniformly on $\mathcal{F} \times \mathcal{R}$. Associate with the boundedness of $f^*$, $f$ and $r$, we obtain that $\mathcal{T}_1 = o_p(1)$ and $\mathcal{T}_2 = o_p(1)$. About $\mathcal{T}^*$, which is an U-statistic, a convergence theorem (Theorem 3.1 in Arcones & Giné (1993)) as $n \to \infty$ implies that the U-statistic converges to $\mathbb{E}_{W, W'|Z}[(f^*(X) - f(X))r(W|Z)(f^*(X') - f(X'))r(W'|Z)]$, where $W' = (Y', X')$ is an i.i.d. copied random element of $W$. Hence, we obtain

$$\mathcal{L}(f, r) = \mathbb{E}_Z\left[\mathbb{E}_{W, W'|Z}[(f^*(X) - f(X))r(W|Z)(f^*(X') - f(X'))r(W'|Z)]\right] + o_p(1)$$
$$= \mathbb{E}_{W, W', Z}[(f^*(X) - f(X))r(W|Z)(f^*(X') - f(X'))r(W'|Z)] + o_p(1),$$

for any $f \in \mathcal{F}$ and $r \in \mathcal{R}$ as $T \to \infty$.

## B.4 ESTIMATION ERROR BOUND OF $\Delta_r$

We can bound $\Delta_r$ as follows:

**Lemma 6.** *Suppose that Assumptions 1 holds. Then,*

$$\Delta_r = \mathcal{O}(\|\hat{r} - r^*\|_{L^2(P \times Q)}) \tag{24}$$

*Proof.* Let us define $\xi_{\hat{f}} := \hat{f} - f^*$. We bound $\Delta_r$ by the Lipschitz continuity of $\mathcal{L}(\hat{f}, r)$ in $r$. For any $r, r' \in \mathcal{R}$, we obtain

$$\mathcal{L}(\hat{f}, r) - \mathcal{L}(\hat{f}, r') = \mathbb{E}\left[\xi_{\hat{f}}^2(X)\xi_{\hat{f}}^2(X')(r(W|Z)r(W'|Z) - r'(W|Z)r'(W'|Z))\right]$$
$$= \mathbb{E}\left[\xi_{\hat{f}}^2(X)\xi_{\hat{f}}^2(X')r(W'|Z)(r(W'|Z) - r'(W'|Z))\right]$$
$$\quad + \mathbb{E}\left[\xi_{\hat{f}}^2(X)\xi_{\hat{f}}^2(X')r'(W|Z)(r(W'|Z) - r'(W'|Z))\right]$$
$$\lesssim 32B'^5\|r - r'\|_{L^2(O)},$$

by the Cauchy-Schwartz inequality and the uniformly bounded property over $\mathcal{F}$ and $\mathcal{R}$. As a result, if the conditional density ratio $\frac{p(w, z)}{p(w)p(z)}$ is bounded, we obtain

$$\Delta_r = \mathcal{O}(\|\hat{r} - r^*\|_{L^2(O)}^2) = \mathcal{O}(\|\hat{r} - r^*\|_{L^2(P \times Q)}^2).$$

$\square$

## B.5 PROOF OF THEOREM 1: ESTIMATION ERROR BOUND

We use the following lemma to show Theorem 1.

**Lemma 7.**

$$\widetilde{\mathcal{L}}_n(\hat{f}, \hat{r}) \leq \Gamma(\hat{f}, \hat{r}) + \mathcal{O}_p(1/n).$$

*Proof.* We start with the following basis inequality following the definition of $\hat{f}$; since it is an minimizer of the empirical risk with $\hat{r}$, we obtain

$$\frac{1}{n}\sum_{j=1}^{n}\left(\frac{1}{n}\sum_{i=1}^{n}(Y_i - \hat{f}(X_i))\hat{r}(W_i|Z_j)\right)^2 \leq \frac{1}{n}\sum_{j=1}^{n}\left(\frac{1}{n}\sum_{i=1}^{n}(Y_i - f^*(X_i))\hat{r}(W_i|Z_j)\right)^2.$$

Since $Y_i = f^*(X_i) + \varepsilon$, the above inequality updated as

$$\frac{1}{n}\sum_{j=1}^{n}\left(\frac{1}{n}\sum_{i=1}^{n}(\varepsilon_i - \xi_{\hat{f}}(X_i))\hat{r}(W_i|Z_j)\right)^2 \leq \frac{1}{n}\sum_{j=1}^{n}\left(\frac{1}{n}\sum_{i=1}^{n}\varepsilon_i\hat{r}(W_i|Z_j)\right)^2, \tag{25}$$

where we define $\xi_{\hat{f}} := \hat{f} - f^*$. Since the left-hand side is expanded as

$$\frac{1}{n} \sum_{j=1}^{n} \left( \frac{1}{n} \sum_{i=1}^{n} (\varepsilon_i - \xi_{\hat{f}}(X_i)) \hat{r}(W_i | Z_j) \right)^2$$

$$= \frac{1}{n} \sum_{j=1}^{n} \left( \frac{1}{n} \sum_{i=1}^{n} \varepsilon_i \hat{r}(W_i | Z_j) \right)^2 + \underbrace{\frac{1}{n} \sum_{j=1}^{n} \left( \frac{1}{n} \sum_{i=1}^{n} \xi_{\hat{f}}(X_i) \hat{r}(W_i | Z_j) \right)^2}_{=\mathcal{L}_n(\hat{f}, \hat{r}) + \mathcal{O}_p(1/n)}$$

$$\underbrace{- 2 \frac{1}{n} \sum_{j=1}^{n} \frac{1}{n^2} \sum_{i,i'=1, i \neq i'}^{n} \varepsilon_i \varepsilon_{i'} \xi_{\hat{f}}(X_i) \xi_{\hat{f}}(X_{i'}) \hat{r}(W_i | Z_j) \hat{r}(W_{i'} | Z_j)}_{=: \Gamma(\hat{f}, \hat{r})} \underbrace{- 2 \frac{1}{n^3} \sum_{i,j=1}^{n} \varepsilon_i^2 \xi_{\hat{f}}(X_i)^2 \hat{r}(W_i | Z_j)^2}_{= \mathcal{O}_p(1/n)}.$$

Substituting this expansion into (25), we obtain

$$\mathcal{L}_n(\hat{f}, \hat{r}) \leq \Gamma(\hat{f}, \hat{r}) + \mathcal{O}_p(1/n) \leq \sup_{f \in \mathcal{F}, r \in \mathcal{R}} \Gamma(f, r) + \mathcal{O}_p(1/n).$$

Therefore,

$$\widetilde{\mathcal{L}}_n(\hat{f}, \hat{r}) = \mathcal{L}_n(\hat{f}, \hat{r}) + \mathcal{O}_p(1/n) = \Gamma(\hat{f}, \hat{r}) + \mathcal{O}_p(1/n) \leq \sup_{f \in \mathcal{F}, r \in \mathcal{R}} \Gamma(f, r) + \mathcal{O}_p(1/n).$$

$\square$

Then we show Theorem 1.

**Step (i): Bound of $\widetilde{\mathcal{L}}_n(\hat{f}, \hat{r})$.**   A goal of this step is to show that the definition of $\hat{f}$ implies that for some $\epsilon \geq 0$, with high probability,

$$\widetilde{\mathcal{L}}_n(\hat{f}, \hat{r}) \leq \epsilon.$$

From Lemma 7,

$$\widetilde{\mathcal{L}}_n(\hat{f}, \hat{r}) \leq \Gamma(\hat{f}, \hat{r}) + \mathcal{O}_p(1/n) \leq \sup_{f \in \mathcal{F}, r \in \mathcal{R}} \Gamma(f, r) + \mathcal{O}_p(1/n).$$

A rest of this step is to bound $\sup_{f \in \mathcal{F}, r \in \mathcal{R}} \Gamma(f, r)$. We bound the tail-probability by the Talagrand's inequality (Theorem 3.3.9 in Giné & Nickl (2021)) for U-statistics (Theorem 5 in Arcones (1995), or Theorem 1 in Li et al. (2014)) as the following inequality; for any $\delta > 0$, with probability at least $1 - \delta$, we obtain

$$\sup_{f \in \mathcal{F}, r \in \mathcal{R}} \Gamma(f, r) \lesssim \mathbb{E}_{\mathcal{D}} \left[ \sup_{f \in \mathcal{F}, r \in \mathcal{R}} \Gamma(f, r) \right] + O \left( \sqrt{\frac{\log(1/\delta)}{n}} \right),$$

provided the uniformly bounded and finite variance properties over $\mathcal{F} \times \mathcal{R}$.

To the end, we study an expectation of $\Gamma(\hat{f}, \hat{r})$ in terms of the dataset $\mathcal{D} = \{(Y_i, X_i, Z_i)\}_{i=1}^{n}$ and obtain

$$\mathbb{E}_{\mathcal{D}} \left[ \sup_{f \in \mathcal{F}, r \in \mathcal{R}} \Gamma(f, r) \right] \leq \mathbb{E}_{\mathcal{D}} [\Gamma(f^*, r^*)] + \mathbb{E} \left[ \sup_{f, f' \in \mathcal{F}, r, f' \in \mathcal{R}} |\Gamma(f, r) - \Gamma(f', r')| \right]$$

$$\lesssim 0 + n^{-1/2} \int_0^B \sqrt{\log \mathcal{N}(\delta, \mathcal{F} \times \mathcal{R}, \| \cdot \|)} d\delta.$$

Here, for the joint set $\mathcal{F} \times \mathcal{R}$, we define a distance $\|(f, r)\| := \|f\| + \|r\|$. The last inequality follows $\Gamma(f^*, r^*) = 0$ and the sub-Gaussianity inequality (Corollary 2.2.8 in van der vaart & Wellner (1996)) associated with the sub-Gaussianity of $\varepsilon_i$ and the Lipschitz continuity of $\Gamma(f, r)$ in $(f, r)$.

By these results, we obtain

$$\mathcal{L}_n(\hat{f}, \hat{r})$$

$$\leq n^{-1/2} \int_0^B \sqrt{\log \mathcal{N}(\delta, \mathcal{F} \times \mathcal{R}, \|\cdot\|)} d\delta + O\left(\sqrt{\frac{\log(1/\delta)}{n}}\right)$$

$$\lesssim n^{-1/2} \int_0^B \sqrt{\log \mathcal{N}(\delta', \mathcal{F}, \|\cdot\|)} d\delta' + n^{-1/2} \int_0^B \sqrt{\log \mathcal{N}(\delta', \mathcal{R}, \|\cdot\|)} d\delta' + O\left(\sqrt{\frac{\log(1/\delta)}{n}}\right),$$

which follows $\log \mathcal{N}(\delta, \mathcal{F} \times \mathcal{R}, \|\cdot\|) \lesssim \log \mathcal{N}(\delta, \mathcal{F}, \|\cdot\|) + \log \mathcal{N}(\delta, \mathcal{R}, \|\cdot\|)$.

**Step (ii): Bound of $\Delta_f$.** To bound $\Delta_f = \widetilde{\mathcal{L}}(\hat{f}, r^*) - \widetilde{\mathcal{L}}_n(\hat{f}, r^*)$, we apply the concentration inequality on empirical processes (displayed in Proposition 3, which is originally developed in Theorem 5 in Arcones (1995)). We regard $h$ in Proposition 3 as $(y, x, y', x') \mapsto (y - f^*(x))(y' - f^*(x'))\Delta_f(x)\Delta_f(x')r(y, x|Z_j)r(y', x'|Z_j)$ and obtain

$$\Delta_f = \mathbb{E}_{W, W', Z}\left[(f^*(X) - f(X))r(W|Z)(f^*(X') - f(X'))r(W'|Z)\right]$$

$$- \frac{1}{n} \sum_{j=1}^n \frac{1}{n(n-1)} \sum_{i,i'=1, i \neq i'}^n (f^*(X_i) - f(X_i))r(W_i|Z_j)(f^*(X_{i'}) - f(X_{i'}))r(W_{i'}|Z_j)$$

$$\lesssim \sqrt{\frac{\log(1/\delta)}{n}},$$

with probability at least $\delta$ for any $\delta \in (0, 1)$.

**Step (iii): Conclusion.** Finally, by combining the above results, with probability at least $1 - \delta$ with $\delta > 0$,

$$\mathcal{L}(\hat{f}, r^*) = \underbrace{\widetilde{\mathcal{L}}(\hat{f}, r^*) - \widetilde{\mathcal{L}}_n(\hat{f}, r^*)}_{=:\Delta_f} + \underbrace{\widetilde{\mathcal{L}}_n(\hat{f}, r^*) - \widetilde{\mathcal{L}}_n(\hat{f}, \hat{r})}_{=:\Delta_r} + \widetilde{\mathcal{L}}_n(\hat{f}, \hat{r})$$

$$\leq \Delta_f + \Delta_r + n^{-1/2} \int_0^B \sqrt{\log \mathcal{N}(\delta', \mathcal{F} \times \mathcal{R}, \|\cdot\|)} d\delta' + O\left(\sqrt{\frac{\log(1/\delta)}{n}}\right)$$

$$\lesssim \mathcal{O}(\|\hat{r} - r^*\|) + n^{-1/2} \int_0^B \sqrt{\log \mathcal{N}(\delta, \mathcal{F}, \|\cdot\|)} d\delta$$

$$+ n^{-1/2} \int_0^B \sqrt{\log \mathcal{N}(\delta, \mathcal{R}, \|\cdot\|)} d\delta + O\left(\sqrt{\frac{\log(1/\delta)}{n}}\right)$$

$$= n^{-1/2} \int_0^B \sqrt{\log \mathcal{N}(\delta, \mathcal{F}, \|\cdot\|)} d\delta + n^{-1/2} \int_0^B \sqrt{\log \mathcal{N}(\delta, \mathcal{R}, \|\cdot\|)} d\delta$$

$$+ O\left(\max\left\{\sqrt{\frac{\log(1/\delta)}{n}}, \frac{1}{n^{1/(2+\gamma)}}\right\}\right) \quad \text{(from Lemma 1).}$$

## C  EXPERIMENTAL SETTINGS AND ADDITIONAL RESULTS

### C.1  SIMULATION STUDIES USING ECONOMICS DATASETS

Here, we report the details of the model and hyperparameters used in our experiments. Since the network structures used in Ai & Chen (2003) and Newey & Powell (2003) have many common features, we describe the network structure in Newey & Powell (2003).

For DeepGMM, DFIV, DeepIV, and KIV, we use the same model and hyperparameters published by Xu et al. (2021a). For IW-LS, we use the following two networks for estimating density ratio and predicting $f^*$ respectively : FC(4,128)-FC(128,128)-FC(128,1) and FC(1,128)-FC(128,128)-FC(128,1). Each fully-connected layer (FC) is followed by leaky ReLU activations with leakiness

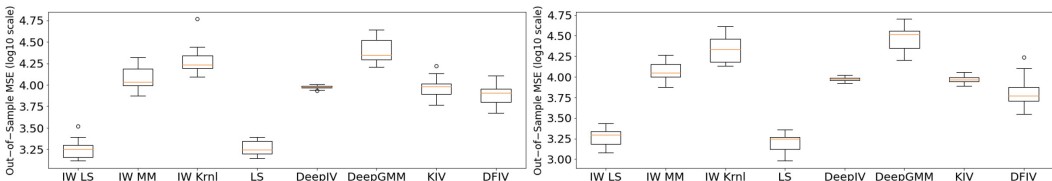

Figure 5: Demand design experiments with $5,000$ samples. The left graph show the results with $\rho = 0.25$ and the right graph shows the results with $\rho = 0.75$.

Figure 6: Demand design experiments with stronger correlation between $X_i$ and $\varepsilon_i$. The sample sizes are $1,000$. The left graph show the results with $\rho = 0.25$ and the right graph shows the results with $\rho = 0.75$.

$\alpha = 0.2$. A regularization coefficient $\eta$ is set to 0.001 as a result of cross-validation. For LS, we use the same network structure in IW-LS for predicting $f^*$. For IW-MM, we use the same network structure as IW-LS. For IW-Krnl, the same network structure as IW-LS to estimate the density ratio and $\zeta$ and $\sigma^2$ are selected via cross-validation. In Ai & Chen (2003) experiment, we change only the first layer in the network structure to match the dimension of $X_i$, and the rest of the network structure is the same.

In addition to the original settings of Newey & Powell (2003) and Ai & Chen (2003), we investigate cases where we add more IVs.

In Newey & Powell (2003), we generate $\{(Y_i, X_i, Z_{i,1}, Z_{i,2}, Z_{i,3}, Z_{i,4})\}_{i=1}^n$, where $Z_{i,2}, Z_{i,3}, Z_{i,4}$ are additional IVs, as follows: first, they generate $\{(\varepsilon_i, U_i, Z_{i,1}, Z_{i,2}, Z_{i,3}, Z_{i,4})\}_{i=1}^n$ from the multivariate normal distribution $\mathcal{N}\left(\begin{pmatrix} 0 \\ 0 \\ 0 \\ 0 \\ 0 \\ 0 \end{pmatrix}, \begin{pmatrix} 1 & 0.5 & 0 & 0 & 0 & 0 \\ 0.5 & 1 & 0 & 0 & 0 & 0 \\ 0 & 0 & 1 & 0 & 0 & 0 \\ 0 & 0 & 0 & 1 & 0 & 0 \\ 0 & 0 & 0 & 0 & 1 & 0 \\ 0 & 0 & 0 & 0 & 0 & 1 \end{pmatrix}\right)$; then, they generate $X_i = Z_{i,1} + Z_{i,2} + Z_{i,3} + Z_{i,4} + U_i$ and $Y_i = f^*(X_i) + \varepsilon_i$, where $f^*(X_i) = \ln(|X_i - 1| + 1)\,\mathrm{sgn}(X_i - 1)$. Here, $\varepsilon_i$ and $U_i$ are unobservable.

In Ai & Chen (2003), they generate $\{(Y_i, X_i, Z_i, W_{i,1}, W_{i,2}, W_{i,3})\}_{i=1}^n$, where $W_{i,1}, W_{i,2}, W_{i,3}$ are additional IVs, as follows: first, we generate $\{(\varepsilon_i, X_{1i}, V_i, U_i)\}_{i=1}^n$ as $\varepsilon_i \sim \mathcal{N}\left(0, X_{1i}^2 + V_i^2\right)$, $X_{1i} \overset{\text{i.i.d.}}{\sim} \mathrm{Unif}[0,1]$, $V_i \overset{\text{i.i.d.}}{\sim} \mathrm{Unif}[0,1]$, $\begin{pmatrix} W_{i,1} \\ W_{i,2} \\ W_{i,3} \end{pmatrix} \overset{\text{i.i.d.}}{\sim} \mathcal{N}\left(\begin{pmatrix} 0 \\ 0 \\ 0 \end{pmatrix}, \begin{pmatrix} 1 & 0.3 & 0.3 \\ 0.3 & 1 & 0.3 \\ 0.3 & 0.3 & 1 \end{pmatrix}\right)$, $W_i = \sum_{j=1}^3 W_{i,j}$, and $U_i \overset{\text{i.i.d.}}{\sim} \mathcal{N}\left(0, X_{1i}^2 + V_i^2 + |W_i|\right)$; second, we generate $X_{2i} = X_{1i} + V_i + W_i + R \times \varepsilon_i + U_i$ and $Y_1 = X_{1i}\gamma_0 + h_0(X_{2i}) + \varepsilon_i$, where $h_0(X_{2i}) = \exp(X_{2i})/(1 + \exp(X_{2i}))$ and $R$ is chosen as 0.9; then, obtain $X_i = (X_{1i}\ X_{2i})^\top$ and $Z_i = (X_{1i}\ V_i\ W_{i,1}\ W_{i,2}\ W_{i,3})$. Here, $\varepsilon_i$ and $U_i$ are unobservable, and $f^*(X_i) = X_{1i}\gamma_0 + h_0(X_{2i})$, where the function $h_0$ and $\gamma_0$ are unknown.

### C.2 SIMULATION STUDIES USING DEMAND DESIGN DATASETS

For DeepGMM, DFIV, DeepIV, and KIV, we use the exact same model and hyperparameters in Xu et al. (2021a). For LS and IW-LS, we use the network structure based on the economics datasets network. $\eta$ is 0.001 as a result of cross-validation. For IW-Krnl, we use the same network structure as IW-LS to estimate the density ratio and select $\zeta$ and $\sigma^2$ via cross-validation.

Figure 7: Demand design experiments with stronger correlation between $X_i$ and $\varepsilon_i$. The sample sizes are $5,000$. The left graph show the results with $\rho = 0.25$ and the right graph shows the results with $\rho = 0.75$.

Figure 5 shows the results with $5000$ samples. First, our proposed IW-LS outperforms the existing methods and minimizes the MSE. Second, LS, which is a naive nonlinear regression without IV, outperforms the other existing methods using IV. In contrast, IW-LS outperforms LS.

We consider the case where endogeneity causes a larger change in outcome. Since the original demand design dataset has a small effect of the bias, we slightly change $V_i$ in the price equation: $P_i = 25 + (C_i + 3)h(T_i) + 10V_i$. In the original dataset, $Y_i$ is generated as $Y_i = 100 + (10 + P_i)S_i h(T_i) - 2P_i + \varepsilon_i$. However, because the impact of $\varepsilon_i$ on the price is very limited, because the variance of $\varepsilon_i$ is relatively small compared to that of $Y_i$. This is the reason why the LS also performs well in the previous result; that is, we can obtain good performance even when ignoring endogeneity (because $f^*(X_i)$) is close to $\mathbb{E}[Y_i|X_i]$). For this reason, this dataset is not appropriate for investigating the performances of the methods, although it is used in existing studies. Here, we consider the different model $Y_i = 100 + (10 + P_i)S_i h(T_i) - 2P_i + 100\varepsilon_i$, in which we multiply the error term by $100$. In this case, the bias has a more serious impact on the values of price and outcome. Figure 6 shows the results with $1,000$ samples, and Figure 7 shows the results with $5,000$ samples. Regardless of the sample size and $\rho$, our proposed method outperforms existing methods. Even in this experiment, the LS still performs well. We consider that this is because the correlation between $X_i$ and $\varepsilon_i$ is not strong. In the dataset, $V_i$, which is a cause of the bias, follows a standard normal distribution and has a limited impact on price due to the constant term $25$ ($P_i = 25 + (C_i + 3)h(T_i) + V_i$), that is, in the variation of $P_i$, $V_i$ has a small effect compared with the other variables.

In this dataset, $f^*(X_i)$ takes large values compared to the error term. Under this situation, a model trained to predict $Y_i$ may perform well because the influence of $\mathbb{E}[\varepsilon_i|X_i] \neq 0$ is limited. However, the purpose of using NPIV in the first place is because the latter influence is large, or else effects of $\mathbb{E}[\varepsilon_i|X_i] \neq 0$ can be ignored. As expected, in our experiments using Hartford et al. (2017), the least-squares method also performs well, even though the training process ignores the problem of NPIV. To investigate the performance for causal inference, we recommend using simpler datasets, before using more complicated datasets.

We also show how the MSE of the IWMM decreases as the sample size grows in Appendix C.4

### C.3 SIMULATION STUDIES USING MNIST DATASETS

We also investigate how our proposed method performs on high-dimensional data. By using MNIST dataset (LeCun & Cortes, 2010), we convert the low dimensional IVs of economics artificial datasets used in Newey & Powell (2003) and Ai & Chen (2003) to high-dimensional IVs. We compare our proposed methods, IW-LS and IW-MM, with the LS and KIV.

**Extension of the dataset in Newey & Powell (2003):** For creating high-dimensional IVs, we equip the IVs used in Newey & Powell (2003) to the feature vector of the MNIST dataset. Let $D_i \in \mathbb{R}^{100}$ be a randomly chosen feature vector of the MNIST dataset of the number 0. We reduce the dimension to $100$ by using the principle component analysis. We multiply the original IV in Newey & Powell (2003) by the feature vector to convert the low-dimensional IV to the high-dimensional IV; that is, we create a new IVs $\tilde{Z}_i \in \mathbb{R}^{100}$ as $\tilde{Z}_i = Z_i D_i$. Instead of $Z_i$, we use this new IV $\tilde{Z}_i$ and estimate the structural function $f^*$. The sample size is $2,500$. The other settings are the same as Section 6.1. The experimental results are shown in the left figure of Figure 8.

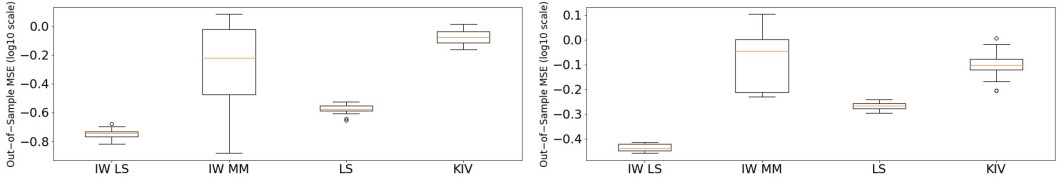

Figure 8: The log10 scaled MSEs using the setting in Newey & Powell (2003). The left graph shows the results using the original dataset with the MNIST dataset. The right graph shows the results with additional IVs and the MNIST dataset.

Figure 9: The log10 scaled MSEs of dataset in Ai & Chen (2003) with the MNIST dataset. The left graph shows the result with $R = 0.1$ and the right graph shows the result with $R = 0.9$.

Next, we add an IV to the original dataset in Newey & Powell (2003) as $\{(Y_i, X_i, Z_{i,1}, Z_{i,2})\}_{i=1}^n \sim$
$$\mathcal{N}\left(\begin{pmatrix} 0 \\ 0 \\ 0 \\ 0 \end{pmatrix}, \begin{pmatrix} 1 & 0.5 & 0 & 0 \\ 0.5 & 1 & 0 & 0 \\ 0 & 0 & 1 & 0 \\ 0 & 0 & 0 & 1 \end{pmatrix}\right),$$ where $Z_{i,2}$ is an additional IV. Then, we generate $X_i = Z_{i,1} +$
$Z_{i,2} + U_i$ and $Y_i = f^*(X_i) + \varepsilon_i$, where $f^*(X_i) = \ln(|X_i - 1| + 1)\operatorname{sgn}(X_i - 1)$. Here, $\varepsilon_i$ and $U_i$ are unobservable. Let $D_{i,1} \in \mathbb{R}^{100}$ be a randomly chosen feature vector of the MNIST dataset of the number 0, and $D_{i,2} \in \mathbb{R}^{100}$ be a randomly chosen feature vector of the MNIST dataset of the number 1. We reduce the dimension to 100 by using the principle component analysis. We multiply the original IVs by the feature vector to convert the low-dimensional IVs to high-dimensional IVs; that is, we create a new IV $\tilde{Z}_{i,1} \in \mathbb{R}^{100}$ as $\tilde{Z}_{i,1} = Z_{i,1}D_{i,1}$ and $\tilde{Z}_{i,2} \in \mathbb{R}^{100}$ as $\tilde{Z}_{i,2} = Z_{i,2}D_{i,2}$. We use these new IVs $\tilde{Z}_{i,1}$ and $\tilde{Z}_{i,2}$ to estimate the structural function $f^*$. The experimental results are shown in the right figure of Figure 8.

**Extension of the dataset in Ai & Chen (2003):** We also equip the MNIST dataset to the IVs of the dataset used in Ai & Chen (2003). Let $D_{i,1} \in \mathbb{R}^{100}$ be a randomly chosen feature vector of the MNIST dataset with the number 0, and $D_{i,2} \in \mathbb{R}^{100}$ be a randomly chosen feature vector of the MNIST dataset with the number 1. We reduce the dimension to 100 by using the principle component analysis. We transform the original IVs in Ai & Chen (2003), $W_{i,1}, W_{i,2} \in \mathbb{R}$, to the high-dimensional IVs by multiplying them with the MNIST feature vectors; that is, we create a new IV $\tilde{W}_{i,1} \in \mathbb{R}^{100}$ as $\tilde{W}_{i,1} = W_{i,1}D_{i,1}$ and $\tilde{W}_{i,2} \in \mathbb{R}^{100}$ as $\tilde{W}_{i,2} = W_{i,2}D_{i,2}$. We use these new IVs, $\tilde{W}_{i,1}$ and $\tilde{W}_{i,2}$, to estimate the structural function $f^*$. The sample size is $2,500$. The other settings are the same as Section 6.1. The experimental results are shown in Figure 9.

### C.4 SIMULATION STUDIES ON THE RATE OF THE MSE

We show experimental results on the MSEs of the NPIV and the classical 2SLS under various sample sizes. Here, we have two goals: (i) we validate the theoretical MSE derived in Section 5.3, and (ii) we show the relative performance of the NPIV method against the classical 2SLS in the learning problem with conditional moment restrictions.

In Figure 10, we show the MSEs of the IWMM in the setting of Newey & Powell (2003) with various sample sizes. We follow the same setting used in Section 6.1, except for the choices of sample sizes. We investigate whether the empirical MSE of IWMM converges to 0 with $\mathcal{O}_p(1/\sqrt{n})$ for sample size $n$, which is theoretically provided in Section 5.3 under some assumptions. We compute the empirical MSEs with sample sizes $\{500, 1000, 1500, 2000, 3500, 4000, 4500, 5000\}$ with 10 trials, then obtain

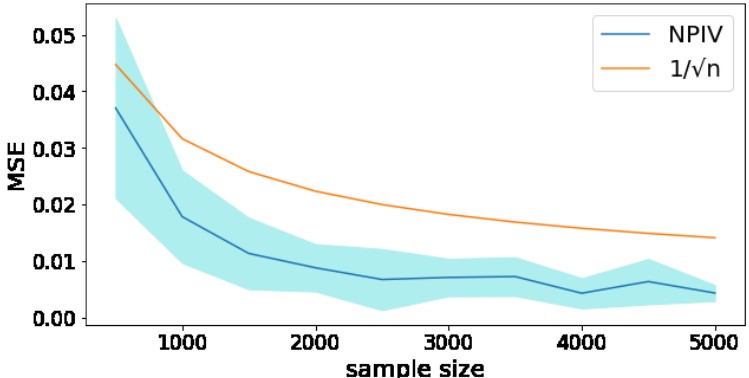

Figure 10: MSEs of the IWMM in the setting of Newey & Powell (2003) with various sample sizes. The light blue region represents the standard deviation.

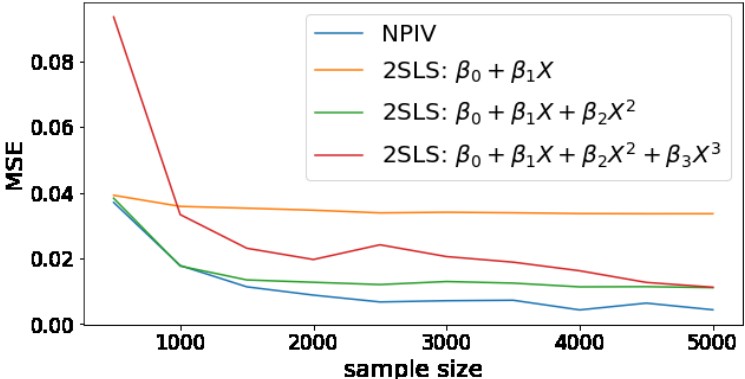

Figure 11: MSE of the IWMM in the setting of Newey & Powell (2003).

its mean and standard deviation. In Figure 10, we compare the empirical MSEs with the line that represents $1/\sqrt{n}$. As in our theoretical results, the MSEs decays in $\mathcal{O}_p(1/\sqrt{n})$.

With the same setting, we also compare the MSEs of IWMM with the classical 2SLS in Figure 11, to understand the effectiveness of the NPIV method. We consider three models for the 2SLS:

$$Y_i = \beta_0 + \beta_1 X_i + \varepsilon_i,$$
$$Y_i = \beta_0 + \beta_1 X_i + \beta_2 X_i^2 + \varepsilon_i,$$
$$Y_i = \beta_0 + \beta_1 X_i + \beta_2 X_i^2 + \beta_3 X_i^3 + \varepsilon_i.$$

The difference between these three models lies in the choice of the polynomial basis. For instance, the last model approximates the structure function by a cubic function. If a model introduces an infinite number of polynomial bases, they can approximate various smooth functions by the series expansion. In other words, as the number of the polynomial basis increases, the classical 2SLS approaches the NPIV. This method of introducing a basis is called sieve regression in econometrics, and Newey & Powell (2003) proposed using it to solve NPIV. The experimental result in Figure 11 also shows that the second and third models, with the polynomial bases, $X_i^2$ and $X_i^3$, perform closer to the NPIV method than the first model using only $X_i$.

It is important to note that the 2SLS with a polynomial basis function is difficult to implement in high-dimensional situations. In the setting of Newey & Powell (2003), because both $X$ and $Z$ have one dimension, polynomial approximation is effective. However, if the dimension increases, the series expansion of multivariate functions requires a huge number of basis functions, hence approximation becomes very difficult. In machine learning, for instance, Singh et al. (2019) proposes to introduce RKHS to avoid this difficulty.

