# OpenReview forum: "Learning Causal Models from Conditional Moment Restrictions by Importance Weighting"
_ICLR.cc/2022/Conference — ICLR 2022 Spotlight_

### Official Review · Reviewer_FAMR · 2021-11-02

**Correctness:** 4
**Technical Novelty And Significance:** 3
**Empirical Novelty And Significance:** 3
**Recommendation:** 8
**Confidence:** 3

**Main Review:**

## Strong points:
- The approach of using importance weighting to overcome the difficulty of learning under conditional moment restrictions is an elegant solution to solve the presented NPIV regression problem.
- The article is well written and the balance between motivation, theory and experiments allows to get a good understanding of the proposal.
- The experiments are well described and motivated. I appreciate the honest assessment of the relevance of the chosen settings for the proposed method.

## Issues/Points that require clarification:
- In Section 3.1, there should be references given for the cited methods (general reference for IV methods, for two-step GMM, CU estimator, etc.)
- The assumptions part is sometimes vague and only phrases certain assumptions without formalizing them. To list a few examples:
  - In Section 2, do you assume i.i.d. samples? If so, please add this when introducing the available dataset and if not, please specify why you don't need i.i.d. samples.
  - In Section 4, the definition of the conditional density ratio function $r^*$ implicitly assumes that $p(y,x)\neq0$ for all $(x,y)\in\mathcal{X}\times \mathbb{R}$. This should be made explicit, and also be discussed w.r.t. the density ratio estimator $\hat{r}$ and maybe also w.r.t. the motivating example. (Your Assumption 2 in Section 5 excludes this case of $p(y,x)=0$ but this should maybe be stated already at the moment where you define $r^*$.)
  - In Section 5, you introduce Assumptions 1 and 2 with the remark "For ease of discussion". Does that mean that these are not necessary?
- Since the authors suggest that their approach should be relevant for learning causal relationships in the context of computer vision and NLP tasks (Section 4.3 and Section 6.2), the claimed added value of their proposal in practice would be more convincing if they could provide an example from either of these contexts where such an IV approach would be suitable.
- At the end of Section 4.3, the authors conjecture that their proposed heuristic works well when $f^*$ is close to $\mathbb{E}[Y_i|X_i]$. If I understood correctly, this means that in the case where $\mathbb{E}[\varepsilon|X_i]\approx 0$ the heuristic should work, but from what I understood this approach is motivated by cases where the assumption $\mathbb{E}[\varepsilon|X_i]= 0$ is clearly violated, motivating the use of IVs. Is that correct or did I misunderstand the heuristic?
- In Lemma 2, what is the role of $T$? From Appendix A.3, it is not clear what $T$ represents. In the proof of Lemma 2, it is written ``_[...] we consider $T$ i.i.d. random variables $W_i=(X_i,Y_i), i=1,\dots,n$ [...]''. What is $T$ in relation to $n$ in this definition? Is it the same $T$ as in Section 5.3?
- Since the authors provide a theoretical bound on the MSE of $f^*$, it would be interesting and a valuable addition to empirically assess this bound.
- Did the authors also assess the performance of the classical linear 2SLS estimator in both experiments (Sections 6.1 and 6.2)? It would be interesting to see the impact of the nonlinear form of the structure function $f^*$ on the performance of this classical estimator to compare with the neural network based approaches.

### Minor comments (that did not impact the score):
- p.1: when learning causal (structural) relationships _are_ $>>$ when learning causal (structural) relationships _is_
- p.1: please add a (standard) reference such as Wooldridge (2010) when you first mention the instrumental variable approach (second paragraph of the introduction).
- p.1: we focus on _on_ nonparametric IV $>>$ we focus on nonparametric IV
- p.2, Section 3.1: the parameter $\theta$ of the linear model: shouldn't it rather be $\theta^*$?
- p.2, Section 3.1: if there are IVs of dimension $d_X$ or more that _satisfies_ $>>$ that _satisfy_
- p.2, Section 3.1: $\tilde{X}=...$: shouldn't it rather be $\hat{X}$?
- Please try to avoid alternating between past and present tense when referring to other works (e.g., p. 3 ``Newey \& Powell (2003) proposed [...] They first define [...]''. I would suggest using only present tense.
- p. 3, Section 3.3: approaches recently _propose_ $>>$ approaches recently _proposed_
- p.5, Section 4.3: as a recent finding, such as Bartlett et al. (2020) in linear regression $>>$ as a recent finding suggests in the case of linear regression (Bartlett et al., 2020).
- p. 6, Section 5: the distribution of $(Y_i, X_i, Z_i$ by $O$ $>>$ the distribution of $(Y_i, X_i, Z_i)$ by $O$

##########################################################################
### Post-rebuttal update

I thank the authors for their detailed, pertinent and timely responses. The reformulated assumptions and other paragraphs, the changed title and the requested additional experimental results have addressed my concerns. I have therefore decided to increase my rating and to recommend this paper to be accepted.

##########################################################################

### References:
[1] Guido W. Imbens and Donald B. Rubin. _Causal inference in statistics, social, and biomedical sciences_. Cambridge University Press, 2015.

[2] Judea Pearl. _Causality_. Cambridge university press, 2009.

[3] Jeffrey Wooldridge. _Econometric Analysis of Cross Section and Panel Data, 2nd edition_. MIT Press, 2010.


**Summary Of The Paper:**

The present paper proposes an importance weighting approach to address the issue of regression under conditional moment restrictions in the context of non-parametric instrumental variables. The authors provide error bounds on the learned structural function. They show that their approach has a convergence rate of $\mathcal{O}(1/\sqrt{n})$.
The main contributions of this work are presented in Sections 4 and 5, where they first introduce their method based on re-weighting by an estimated conditional density ratio function and then provide an estimation error analysis.
The theoretical results are complemented with two synthetic experiments which show that the proposed method can compete with and in certain cases improves upon state of the art.

**Summary Of The Review:**

In summary, I am rather convinced that the contribution of this paper is important and its theoretical as well as empirical findings are well described and motivated. However the authors should try to provide relevant applications or settings for their method that would indeed benefit from this novelty (compared to existing works). The experiments do not support the authors' claim that their method outperforms other competitors. Also, they should maybe clarify the paper's content in the title by specifying the context of IVs instead of ``learning causal relationships''. Indeed, from the title one expects to find references to the classical causal inference frameworks (either Imbens \& Rubin (2015) or Pearl (2009)) and specification of causal estimands.
I will read the rebuttal carefully and am willing to increase the score if the authors address the raised concerns.

---

> ### Author Response · Authors · 2021-11-15
> **Response to Reviewer FAMR**
>
> We greatly appreciate your constructive comments. We revised our manuscript based on your advice and continue to reflect your comments on our manuscript.
>
> Our replies to the issues you raised are listed below.
>
> **Q1.**
> > In Section 3.1, there should be references given for the cited methods...
>
> **A1**
> Thank you for your suggestion. In the revised manuscript, we added the citations to these methods.
>
> ------------
> **Q2**
> > The assumptions part is sometimes vague and only phrases certain assumptions without formalizing them. To list a few examples:
>
> Thank you for pointing this out. We addressed the points you raised as follows.
>
> ------------
> **Q2.1**
> > In Section 2, do you assume i.i.d. samples? If so, please add this ...
>
> **A2.1**
> We assume that the observations are i.i.d. In the revised manuscript, we added the assumption in Section 2.
>
> ------------
>
> **Q2.2**
> > In Section 4, the definition of the conditional density ratio function $r^*$ implicitly assumes that $p(y,x) \neq 0$ for all $(x,y) \in \mathcal{X} \times \mathbb{R}$. This should be made explicit, and also be discussed w.r.t. the density ratio estimator $\hat{r}$ and maybe also w.r.t. the motivating example. (Your Assumption 2 in Section 5 excludes this case of $p(y,x) = 0$ but this should maybe be stated already at the moment where you define $r^*$.)
>
> **A2.2**
> As you point out, we assume that $p(y, x) > 0$ for all $(y,x)\in\mathcal{Y}\times \mathcal{X}$. Note that we have already assumed the existence of $p(y, x| z)$ for all $(y,x,z)\in\mathcal{Y}\times \mathcal{X}\times \mathcal{Z}$ by assuming the existence of the conditional moment restrictions. We have clarified them in the revised manuscript. Additionally, we restate the associated assumption on the conditional density ratio function $r(x) = \frac{p(y, x| z)}{p(y, x)} = \frac{p(y, x, z)}{p(y, x)p(z)}$ at the top of Section 4.
>
> The assumption that $p(y, x) > 0$ for all $(y,x)\in\mathcal{Y}\times \mathcal{X}$ is standard in density ratio estimation to define the density ratio. See Sugiyma et al. (2012)
>
> Masashi Sugiyama, Taiji Suzuki, and Takafumi Kanamori. Density Ratio Estimation in MachineLearning, 2012.
>
> ------------
>
> **Q2.3**
> > ...you introduce Assumptions 1 and 2 with the remark "For ease of discussion". ... these are not necessary?
>
> **A2.3**
> We mentioned it because there is a possibility that it can be extended to distributions other than sub-Gaussian distribution in some situations. However, since it is misleading, we have removed it in the revised manuscript.
>
> ------------
>
> **Q3**
> > Since the authors suggest that their approach should be relevant ... in the context of computer vision and NLP tasks, the claimed added value of their proposal in practice would be more convincing if they could provide an example...
>
> **A3**
> Thank you for your suggestion. In the revised manuscript, we have cited relevant papers.
>
> ------------
>
> **Q4**
> > At the end of Section 4.3, the authors conjecture that their proposed heuristic works well when $f^*$ is close to $E[Y_i | X_i]$. If I understood correctly, this means that in the case where  the heuristic should work, but from what I understood this approach is motivated by cases $E[Y_i | X_i] \neq 0$ where the assumption $E[Y_i | X_i] = 0$ is clearly violated, motivating the use of IVs. Is that correct or did I misunderstand the heuristic?
>
> **A5**
> The heuristic does not assume $\mathbb{E}[\varepsilon_i| X_i]=0$. The motivation is to select a function $f(x)$ that is the closest to $\mathbb{E}[Y_i| X_i=x]$ among functions satisfying the conditional moment restriction.
> This heuristic works well when $f^*(x)$ takes a near value of $\mathbb{E}[Y_i| X_i=x]$ while $f^*(x)\neq \mathbb{E}[Y_i| X_i=x]$.
>
> When there are multiple functions that satisfy the moment constraint, choosing the function that is better in terms of prediction is an open problem. For instance, see Zhang et al. (2021). We explain this in the revised manuscript.
>
> Rui Zhang, Krikamol Muandet, Bernhard Schölkopf, Masaaki Imaizumi. Instrument Space Selection for Kernel Maximum Moment Restriction, 2021.
>
> ------------
>
> **Q6**
> > In Lemma 2, what is the role of $T$? ... Is it the same $T$ as in Section 5.3?
>
> **A6**
> They are typos. Thank you for pointing them out. In the revised manuscript, we corrected the first $T$ to $n$ and removed the second $T$.
>
> ---------
>
> **Q7**
> > it would be interesting and a valuable addition to empirically assess the bound of $f^*$.
>
> > ...see the impact of the nonlinear form of the structure function on the performance of this classical estimator...
>
> **A7**
> Thank you for your suggestion. For the first suggestion, we will add an experiment on the estimation error for a simple dataset. We will check if the estimation error decreases with the same order theoretically shown by us. For the second suggestion, we will also investigate the performances for a simple dataset and standard estimators. We will show them during the rebuttal period or in the camera-ready.

---

> > ### Comment · Reviewer_FAMR · 2021-11-18
> > **Thank you for your detailed answers and revisions**
> >
> > I thank the authors for taking the time to answer my questions and addressing the raised issues point by point.
> >
> > The authors have indeed addressed most of the issues by clarifying and refining their statements.
> > I am rather inclined to change my score, however I would like to first see the requested additions in the experiments section. One of the requested/suggested additions is the assessment of the classical linear 2SLS estimator. From my experience with this estimator and given your experimental settings, it should be relatively easy to apply this estimator on your proposed simulations. The same holds for the estimation error experiment that the authors have already announced in their response to my review.
> > I would also like to repeat my recommendation from my "Summary of the review": since the manuscript focuses on nonparametric IV regressions and not so much on causal effect estimation or causal discovery, I would suggest changing the title to specify the context of IVs instead of simply stating "learning causal relationships". Or if the authors disagree with this opinion, I would like to know their argument for this choice.
> >
> > Two general comments:
> >
> > - For future revisions (at ICLR or other venues) I would encourage the authors to highlight the changes in the revised manuscript in a different colour to allow for a better re-assessment of their revised manuscript.
> > - In the response to reviewer ThQb the authors mention additional changes they plan to do using the additional page for the camera-ready. From my understanding, the number of pages stays the same between the initial submission and the camera-ready: "_There will be a strict upper limit of 9 pages for the main text of the submission, with unlimited additional pages for citations. This page limit applies to both the initial and final camera ready version._" (from https://iclr.cc/Conferences/2022/CallForPapers)

---

> > > ### Author Response · Authors · 2021-11-20
> > > **Re: Thank you for your detailed answers and revisions**
> > >
> > > Thank you very much for your reply.
> > >
> > > > For future revisions (at ICLR or other venues) I would encourage the authors to highlight the changes in the revised manuscript in a different colour...
> > > > In the response to reviewer ThQb the authors mention additional changes they plan to do using the additional page for the camera-ready. From my understanding, the number of pages stays the same ...
> > >
> > > We greatly thank you again for your advice. First, we did not know that we were allowed to change the text color during the rebuttal stage. From now on, we will highlight sentences that have been changed using a different color. Second, based on our experience at other conferences, we mistakenly assumed that a page would be added in the camera-ready. We will explain our misunderstanding to Reviewer ThQb. To address his concern, we will add another section in the appendix to provide further explanations for readers unfamiliar with this subject, as well as Dikkala et al. (2020).
> > >
> > > We will shortly post a comment addressing the whole group, summarizing the changes and our plan for the camera-ready.
> > >
> > > Our replies to the other comments are listed below.
> > >
> > > **Q1**
> > > One of the requested/suggested additions is the assessment of the classical linear 2SLS estimator. From my experience with this estimator and given your experimental settings, it should be relatively easy to apply this estimator on your proposed simulations. The same holds for the estimation error experiment that the authors have already announced in their response to my review.
> > >
> > > **A1**
> > > In Appendix B.3 of the revised manuscript, we have added experiments on the convergence rate of MSE and the comparison between NPIV and 2SLS, using the setting of Newey and Powell (2003). First, we can confirm that the MSE of our proposed method decreases with order close to $O(1/\sqrt{n})$, which is expected from the theoretical results. Second, we compared our proposed method with three linear models estimated by the 2SLS, which are $Y = \beta_0 + \beta_1X$, $Y = \beta_0 + \beta_1X + \beta_2X^2$, and $Y = \beta_0 + \beta_1X^1 + \beta_2X^2 + \beta_3X^3$. As we increase the polynomial term, the performance improves. This is because adding infinite　polynomial terms corresponds to the Taylor expansion, approximating a nonparametric structural function. These linear-in-parameter models are called sieve regression models, mainly in economics, and studied by Newey and Powell (2003) for NPIV.
> > >
> > > Our simulation studies show that our proposed method is appropriate in the setting of NPIV. Although we can improve the performance of the classical 2SLS by increasing the polynomial term (nonparametric regression by sieve regression), it is prohibitively difficult when the dimension of the dataset is large, due to considering the Taylor expansion of the polynomial function.
> > >
> > > We checked the points raised by the reviewer with a very simple setup. We will also add at least one other experimental setting in the camera-ready. If the reviewers have any suggestions or ideas on another suitable settings, we will follow it and add to the camera-ready.
> > >
> > > **Q2**
> > > I would also like to repeat my recommendation from my "Summary of the review": since the manuscript focuses on nonparametric IV regressions and not so much on causal effect estimation or causal discovery, I would suggest changing the title to specify the context of IVs instead of simply stating "learning causal relationships". ..
> > >
> > > **A2**
> > > The reason why we used "Learning causal relationships" is because the proposed method allows us to learn causal models under more general conditional constraints. This is similar to the difference between the formulations of Newey and Powell (2003) and Ai and Chen (2003). Therefore, although NPIV is the most representative application (thus our focus in this paper), we use "Learning causal relationships" as the title because it allows us to estimate a broader class of models defined generally by conditional moment constraints.
> > >
> > > However, as you point out, we agree that the word "relationships," is somewhat misleading, giving the impression that the paper is about other methods, such as causal discovery. Therefore, we agree that the title should be changed, and have come up with two alternatives:
> > > "Learning Causal Models with Conditional Moment Restrictions by Importance Weighting"
> > > or
> > > "Learning Causal Models with Conditional Moment Restrictions by Importance Weighting: Application to Instrumental Variable Estimation."
> > > We think both are more adequate and better represents our paper compared to the previous title, with the latter being more specific (though considerably longer). Specifically, the term "models" has been used in the literature for this context (see, e.g., Ai and Chen (2003), "Efficient Estimation of Models with Conditional Moment Restrictions Containing Unknown Functions" ). If you have a preference for one or the other, please let us know, and we will modify our title based on your recommendation.

---

> > > > ### Comment · Reviewer_FAMR · 2021-11-21
> > > > **Final remarks before end of discussion period**
> > > >
> > > > I thank the authors for their quick and concise answers to my last comment. The added simulations checking the convergence rate of the proposed method and comparing to 2SLS are an important addition in my opinion and help the reader in further assessing the proposed method.
> > > >
> > > > Concerning the title, I thank the authors for taking the time to propose alternatives to respond to my raised concern about the link with other causal methods. I would tend to choose the second title as I think that it could potentially draw attention more easily of people interested in instrumental variables. But given the length of this second title, I would also understand the choice of this shorter version.

---

> > > > > ### Author Response · Authors · 2021-11-22
> > > > > **Re: Final remarks before end of discussion period**
> > > > >
> > > > > Thank you very much for your comments. We also believe that the added experiments will clarify the contributions of our manuscript to the community.
> > > > >
> > > > > Thank you also for your constructive comments on the title. Then, we use a shorter version. We have updated the title in the manuscript as  "Learning Causal Models with Conditional Moment Restrictions by Importance Weighting." However, we will continue to think about putting "nonparametric instrumental variable methods" in the title for researchers who are interested in nonparametric instrumental variable methods.

---

### Official Review · Reviewer_ThQb · 2021-11-03

**Correctness:** 4
**Technical Novelty And Significance:** 3
**Empirical Novelty And Significance:** 3
**Recommendation:** 8
**Confidence:** 2

**Main Review:**

Improving non-parametric instrumental variables methods is an important problem in causal effect estimation. Based on my limited knowledge, I think this paper is providing a very interesting and promising result, and I do appreciate the theoretical analysis of the estimation error.

My only concern is that it is a bit technical and hard to follow for the general ICLR community, so I would suggest the authors to try to make it a bit more accessible and readable and therefore even more impactful.

In particular, as a minor point, I think the abstract is a bit too vague and would benefit from a more precise description of what the paper is about.

**Summary Of The Paper:**

The paper presents a method for estimating causal effects (or more generally non/parametric functions) under conditional moment restrictions, focusing in this case on nonparametric IVs. The main idea is casting these conditional restrictions to an unconditional version through importance resampling using a conditional density estimator (e.g. based on a least-squares method with a NN). The paper also provides a theoretical analysis of the estimation error, providing am error bound (Thm1). The method is evaluated based on three econometrics datasets from literature, showing a smaller MSE than the compared methods.

**Summary Of The Review:**

The paper seems to provide a novel, interesting and applicable method that improves the accuracy of an important causal inference tool (instrumental variables).

---

> ### Author Response · Authors · 2021-11-15
> **Response to Reviewer ThQb**
>
> Thank you for your comments and positive feedback!
>
> **Q**
> My only concern is that it is a bit technical and hard to follow for the general ICLR community, so I would suggest the authors to try to make it a bit more accessible and readable and therefore even more impactful.
>
> In particular, as a minor point, I think the abstract is a bit too vague and would benefit from a more precise description of what the paper is about.
>
> **A**
> In the previous manuscript, we tried to explain, as clearly as possible given the strict page limit, the problem setting, existing literature, and our proposed method, assuming that readers may not be familiar with the field. In the updated manuscript, we have revised the abstract to better describe the contributions of the paper. If the paper is accepted, we will use the additional page to expound on the abstract, problem setting, and our method further.

---

### Official Review · Reviewer_S3Qv · 2021-11-04

**Correctness:** 3
**Technical Novelty And Significance:** 3
**Empirical Novelty And Significance:** 2
**Recommendation:** 6
**Confidence:** 3

**Main Review:**

Instrumental variable regression is an important problem to tackle for a number of application fields. Therefore an improvement of the methodology in this area should be welcome for the conference audience.

The paper presents a method that is novel in terms of the estimand it proposes, the density ratio, in order to learn the structural function. Their method can be used with various models and objective functions in learning the causal function. The authors present error analyses for the density ratio and the causal function, as well as experiment results comparing their work to recent algorithms in the field. These constitute the strengths of the present work.

One of the weaknesses of the present work is in its presentation. Which shortcomings of the original methods are addressed by the present work, and how, is never clearly presented. The lack of description regarding this issue leads the reader to look for a clear performance increase from the present work.

This leads me to the second weakness of the paper, which is that the experiments do not demonstrate a unanimous advantage for a specific method proposed by the authors. The (potential) advantages of the paper could be more accurately depicted by extending the experiments, e.g. as in Xu et al. (2021).

Though I recommend the acceptance of the paper, I think that the paper would benefit from the aforementioned issues being addressed.

**Summary Of The Paper:**

The authors propose a novel method for conducting instrumental variable regression by learning the ratio of conditional and unconditional densities, and using this estimate when learning the causal relationship between cause and effect variables. The authors present error analyses for their estimation tasks, as well as empirical comparison to previous works.

**Summary Of The Review:**

The authors present a novel method for instrumental variable regression that promises to be useful in various applications. However the motivation for the method and its empirical testing should be improved.

---

> ### Author Response · Authors · 2021-11-15
> **Response to Reviewer S3Qv**
>
> Thank you for your insightful comments. We updated our manuscript based on your feedback.
>
> Our replies to your comments are listed as follows.
>
> **Q1**
> One of the weaknesses of the present work is in its presentation. Which shortcomings of the original methods are addressed by the present work, and how, is never clearly presented. The lack of description regarding this issue leads the reader to look for a clear performance increase from the present work.
>
> **A1**
> The main advantage of our proposed method is its flexibility; being able to use appropriate regression methods for high-dimensional, or some complex data.
> Classical methods, such as Newey and Powell (2003) and Ai and Chen (2003), approximate the nonparametric structural function or conditional moment restriction by sieve regression models. The sieve regression model is known to be difficult to use in terms of predictive performance and computation when the data is high dimensional.
>
> In addition, our method has advantages over recent machine learning methods.
> Recent methods incorporate machine learning techniques in order to handle more complex problems. However, some of those methods rely on computationally difficult methods such as resampling and minimax optimization. For this problem, our method is based on two-step estimation, which is straightforward and easy to implement.
>
> Finally, as in Ai and Chen (2003), our method is also able to solve a broader class of problems, not only NPIV. We describe these advantages on page 5 in the revised manuscript.
>
> **Q2**
> This leads me to the second weakness of the paper, which is that the experiments do not demonstrate a unanimous advantage for a specific method proposed by the authors. The (potential) advantages of the paper could be more accurately depicted by extending the experiments, e.g. as in Xu et al. (2021).
>
> **A2**
> Thank you for your constructive suggestion. We agree that we can improve the presentation of this paper by showing more experiments with a wider range of classes. We will add such results during this rebuttal period or in camera-ready.

---

> > ### Comment · Reviewer_S3Qv · 2021-11-29
> > **Thank you for your response**
> >
> > I thank the authors for their response. I think the conference audience would benefit from the contributions of the paper, and thus recommend its acceptance.

---

### Author Response · Authors · 2021-11-22
**To All Reviewers**

Thank you to all reviewers for your constructive comments. In particular, we are grateful to Reviewer FAMR for the feedback.

First of all, we apologize to Reviewer ThQb for our inaccurate comments. We expected a page to be added in the camera-ready, but this is not the case as pointed out by Reviewer FAMR. As an alternative, we will add supplementary material to the Appendix, similar to Dikkala et al. (2020).

We have updated the manuscript in response to your suggestions. The most significant changes are as follows (we colored them red):
1. In Sections 2 and 4.1, we clarified our assumptions about the data generation process and the density ratio.
2. In Section 4.2, we summarized the advantages of our method.
3. In Appendix B.3, we confirmed the empirical convergence rate of MSE and compared our proposed method with the classical 2SLS through numerical experiments.

We will also do the following in the camera-ready:
1. Add supplementary material to the Appendix.
2. Add experiments using image datasets, as in Xu et al. (2021).
3. Add more experiments and discussions on the comparison of NPIV and 2SLS.

---

### Decision · Program_Chairs · 2022-01-20

**Decision:**

Accept (Spotlight)

**Comment:**

This paper proposes a method that uses conditional moment restriction methods to estimate causal parameters in non-parametric instrumental variable settings.  This is done by converting to an unconditional moment restriction setting common in the econometrics causal inference literature.

The paper was reviewed quite favorably by reviewers, and the authors updated the manuscript to address specific issues raised by reviewers.

---

> ### Public Comment · ~Masahiro_Kato1 · 2022-03-14
> **On the paper title**
>
> Dear Program Chairs,
>
> Thank you for the positive feedback. We also thank everyone involved in the peer-review process.
>
> We have a question. Reviewer FAMR suggested changing the title of our paper, and we agreed. However, it seems that we cannot change the title on OpenReview. Can we change the title to reflect the review comments? If it is not possible to change the title on the OpenReview, we will keep the initial title at the time of submission.